# TOWARDS FEDERATED RLHF WITH AGGREGATED CLIENT PREFERENCE FOR LLMS

**Feijie Wu**[1]**, Xiaoze Liu**[1]**, Haoyu Wang**[2]**, Xingchen Wang**[1]**, Lu Su**[1]**, Jing Gao**[1]
[1]Purdue University    [2]State University of New York at Albany
{wu1977, xiaoze, wang2930, lusu, jinggao}@purdue.edu
hwang28@albany.edu

## ABSTRACT

Reinforcement learning with human feedback (RLHF) fine-tunes a pretrained large language model (LLM) using user preference data, enabling it to generate content aligned with human preferences. However, due to privacy concerns, users may be reluctant to share sensitive preference data. To address this, we propose utilizing Federated Learning (FL) techniques, allowing large-scale preference collection from diverse real-world users without requiring them to transmit data to a central server. Our federated RLHF methods (i.e., FedBis and FedBiscuit) encode each client's preferences into binary selectors and aggregate them to capture common preferences. In particular, FedBiscuit overcomes key challenges, such as preference heterogeneity and reward hacking, through innovative solutions like grouping clients with similar preferences to reduce heterogeneity and using multiple binary selectors to enhance LLM output quality. To evaluate the performance of the proposed methods, we establish the first federated RLHF benchmark with a heterogeneous human preference dataset. Experimental results show that by integrating the LLM with aggregated client preferences, FedBis and FedBiscuit significantly enhance the professionalism and readability of the generated content.

⦿  https://github.com/HarliWu/FedBiscuit

## 1 INTRODUCTION

Large language models (LLMs) exhibit broad knowledge coverage as they are pretrained on a large corpus (Zhao et al., 2023a; Singhal et al., 2023; Wang et al., 2023a; Liu et al., 2024a). To elevate the quality of their generated content to match the professionalism and readability of human writing, a common approach is to fine-tune these models using reinforcement learning with human feedback (RLHF) (Ziegler et al., 2019; Christiano et al., 2017; Ouyang et al., 2022). This process relies on preference datasets, which are constructed through two main methods: human evaluation (Bai et al., 2022; Ganguli et al., 2022; Stiennon et al., 2020) and ChatGPT-based ranking (Dubois et al., 2024b). In the human-effort approach, a set of instructions (a.k.a. prompts or user queries) is paired with multiple model completions (a.k.a. generated responses), and a team of labelers ranks completions of each instruction from best to worst. In the ChatGPT-based method, the dataset is developed by feeding ChatGPT with multiple inputs encompassing an instruction and a pair of completions, where it selects the superior completion for each input.

As LLMs are deployed to serve diverse users, there could be a gap between the preferences of real-world users and those of lablers/ChatGPT, hindering the LLM's ability to generate responses that align with the users' preferences. Therefore, there is a need for a preference dataset that *accurately reflects real world users' preferences in order to enhance the ability of LLM in content generation.*

One straightforward approach to meeting this need is to directly collect preference data from a large number of real-world users and build a comprehensive preference dataset, which can be used to fine-tune an LLM on a central server. This strategy has been implemented in recent projects like OASST (Köpf et al., 2024). However, this approach is

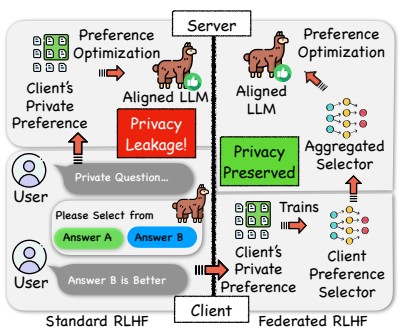

Figure 1: Comparison between standard and federated RLHF

often impractical because it involves collecting LLM users' inputs and preferences – both highly private and sensitive (shown in the left part of Figure 1). Most users are unwilling to share their data, and such practices may also violate regulations like the GDPR (2016) and CCPA (2023), which prohibit the use of users' private data for model training without explicit consent.

To address privacy concerns, we propose *utilizing Federated Learning (FL) techniques to enable large-scale preference collection from diverse real-world users without requiring them to transmit their preference data to a central server.* In our design, we adopt FedAvg, a well-established FL algorithm (Konečn'y et al., 2016; McMahan et al., 2017), to learn user preferences without directly collecting their data. In line with the reward model training used in RLHF (Stiennon et al., 2020; Ouyang et al., 2022; Rafailov et al., 2023), each LLM user works as a FL client and locally trains a reward model that can evaluate the quality of model completions. The server then aggregates these client-trained reward models into a global model. While this approach appears effective, we observe the following two limitations during training:

- **Excessive Computation Overhead:** Typically, the reward model is a regression model designed to output a scalar value that represents the quality of a model completion (Stiennon et al., 2020). Its optimization is based on comparing reward differences between preferred and dispreferred completions, where preferred completions should receive higher rewards. However, during optimization, each data sample requires the retention of two computation graphs (one for the preferred and one for the dispreferred completion) throughout the forward and backward passes. This results in significant computational overhead and heavy GPU demands.
- **Preference Heterogeneity and Reward Hacking:** Client preferences are heterogeneous, as both instructions and preferences vary across clients. As a result, each client tends to train their reward model toward a local minimum, which deviates from the global optimum, leading to longer convergence times compared to centralized training. Moreover, fine-tuning the pretrained model using the reward model can lead to overfitting, where the proxy reward (the reward model's output) improves while the actual performance worsens. This phenomenon, known as reward hacking or reward overfitting, has been discovered in several studies (Askell et al., 2021; Michaud et al., 2020; Tien et al., 2022; Skalse et al., 2022).

In this paper, we propose to address these two limitations and propose effective and computationally efficient methods for preference collection and subsequent fine-tuning. We start with a solution that addresses computation costs. The key idea is to train a binary selector that identifies the superior response between two model completions. Compared with traditional reward model training, the binary selector requires significantly less computation. Casting binary selector training into a federated learning setting, we develop a federated binary selector training (FedBis) framework, as depicted in the right part of Figure 1. The aggregated binary selector captures the common preferences of a large group of users, and thus can simulate a comprehensive preference dataset, facilitating RLHF approaches (e.g., DPO (Rafailov et al., 2023)) to fine-tune the LLM without the need for the real client preference data.

To further address the performance deterioration due to preference heterogeneity and reward hacking, we propose a method named FedBis with cluster-wise aggregation (FedBiscuit). This approach ensembles multiple binary selectors, each trained by clients with similar preferences. Since privacy concerns prevent the explicit sharing of client data, the server intermittently collects the training loss of all binary selectors on the clients. Using this data, clients are grouped into disjoint clusters, and when comparing two completions, the one favored by the majority of selectors is deemed superior. This method has two main advantages: (1) Clients with similar preferences collaboratively train a binary selector, reducing data heterogeneity and improving performance stability, and (2) Reward hacking is mitigated by employing multiple binary selectors, as it becomes difficult for the LLM to generate content that satisfies all binary selectors without making genuine improvements.

**Contributions.** In this paper, our contributions are highlighted as follows:

- To the best of our knowledge, this is the first work to employ federated learning technique to enable large-scale user preference collection for RLHF without jeopardizing user privacy. Our proposed federated RLHF model (i.e., FedBis) encodes each client's preference information into a binary selector and aggregates all clients' binary selectors to capture their common preferences. By aligning the LLM with the aggregated client preference, we can improve the professionalism and readability of LLM's generated content.

- We identify the inherent challenges of federated RLHF, such as preference heterogeneity and reward hacking, and extend FedBis into FedBiscuit with innovative solutions to address these challenges, including grouping the binary selectors of clients with similar preferences to reduce data heterogeneity and employing multiple binary selectors to force the LLM to improve the quality of its generated content.
- We conduct extensive experiments to evaluate the performance of the proposed FedBis and FedBiscuit. Since no prior work has addressed RLHF in a federated learning setting, we establish the first FL benchmark by creating a heterogeneous human preference dataset. As expected, both FedBis and FedBiscuit show significant performance improvements over the base models, Gemma and LLaMA.

## 2 RELATED WORK

**Federated Fine-Tuning for LLM.** Recent studies have increasingly focused on fine-tuning large language models (LLMs) using federated datasets (Sun et al., 2024; Zhang et al., 2024; 2023; Yi et al., 2023; Qin et al., 2023; 2024; Bai et al., 2024). However, these approaches often suffer from high computation and communication costs due to the necessity of training and synchronizing the model with clients. To mitigate these issues, lightweight methods such as black-box fine-tuning (Sun et al., 2023; Lin et al., 2023) and offsite-tuning (Wu et al., 2024a; Kuang et al., 2023) have emerged. Despite their advancements, these methods primarily focus on fine-tuning LLMs for specific downstream tasks, neglecting user preferences in the generated responses. A recent benchmark, OpenFedLLM (Ye et al., 2024), introduces FedDPO, which allows federated clients to optimize their local LLMs using DPO loss. While this approach can potentially align LLMs with human preferences, it faces three key challenges: excessive computational overhead, preference heterogeneity, and the risk of reward hacking. To address these limitations, our work aims to enable LLMs alignment with a feasible and sustainable training framework in FL.

**Reinforcement Learning with Human Feedback (RLHF).** RLHF typically involves supervised fine-tuning, reward modeling, and reward optimization, initially proposed by Christiano et al. (2017). Proximal Policy Optimization (PPO) (Schulman et al., 2017) is a common RLHF algorithm, yet it struggles with instability, inefficiency, and high resource demands (Choshen et al., 2019; Engstrom et al., 2020). These challenges have led to the development of alternative methods, such as Direct Preference Optimization (DPO) (Rafailov et al., 2023) and others (Dong et al., 2023; Zhao et al., 2023b; Azar et al., 2024; Ethayarajh et al., 2024; Gulcehre et al., 2023), which offer more stable and efficient solutions. However, these methods typically operate within a centralized training framework, where the LLM owner retains control over the preference data. In contrast, our work aims to expand data sources and integrate real user preferences without directly collecting their personal data.

## 3 PRELIMINARY

### 3.1 FEDERATED LEARNING (FL)

FL is a distributed training paradigm where a server coordinates various clients toward the same goal, i.e., training a generalized model $\phi \in \mathbb{R}^d$ (Konečn'y et al., 2016; McMahan et al., 2017). Consider an FL system with $M$ clients. Denote the weight of client $m$ as $p_m$ such that $\sum_{m \in [M]} p_m = 1$, and we aim to optimize the following objectives:

$$\min_{\phi \in \mathbb{R}^d} \quad F(\phi) \stackrel{\triangle}{=} \sum_{m \in [M]} p_m F_m(\phi), \tag{1}$$

where $F_m(\phi)$ is the expected loss on client $m$ given the model $\phi$. As a classical FL algorithm, FedAvg can solve the optimization problem by multiple communications between the server and the clients, followed by a number of recent works (Wang et al., 2021; 2022; 2023c; 2024b;c; Zhang et al., 2021). In each communication round $r \in [R]$ with the global model $\phi_r$, the following steps are conducted:

- **Model broadcast:** The server uniformly samples $A$ clients without replacement, denoted by $\mathcal{A}$, broadcasts the global model $\phi_r$ to the sampled clients.
- **Local training on client** $m \in \mathcal{A}$**:** The client initializes the local model $\phi_{r,0}^m$ with the received $\phi_r$. In the next $K$ iterations, the client updates its local model via $\phi_{r,k+1}^m = \phi_{r,k}^m - \eta \nabla F_m(\phi_{r,k}^m), k \in [K]$, where $\eta$ is learning rate, and $\nabla F_m(\phi_{r,k}^m)$ is the local gradient on $\phi_{r,k}^m$ and estimated by a mini-batch. We denote this $K$-iteration local training by $\phi_{r,K}^m = \text{OPTIM}(m, \phi_r, K)$, which optimizes the model $\phi_r$ with the data of client $m$.

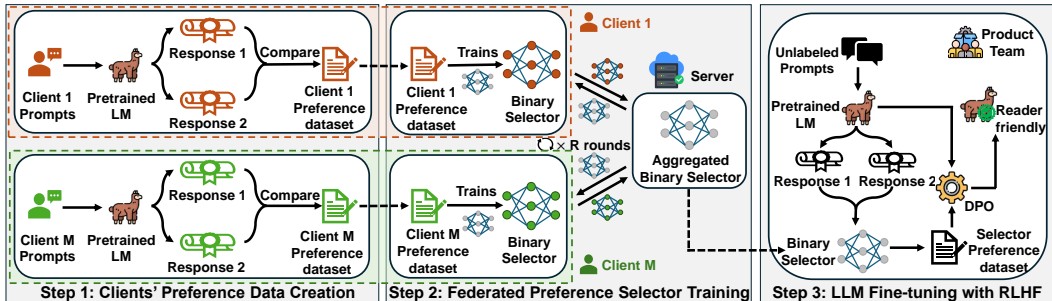

Figure 2: An outline of the proposed FedBis, an RLHF method in federated learning.

- **Global aggregation:** The server collects the local model $\phi^m_{r,K}$ from clients $m \in \mathcal{A}$ and updates the global model via weighted average aggregation, i.e., $\phi_{r+1} = \frac{M}{A} \sum_{m \in \mathcal{A}} p_m \phi^m_{r,K}$.

### 3.2 REINFORCEMENT LEARNING WITH HUMAN FEEDBACK (RLHF)

The objective of RLHF is to align a pretrained language model with human preferences so that the model can generate text that is as professional and readable as human writing. DPO (Rafailov et al., 2023) is one of the most effective ways to achieve the goal. There is a human-annotated dataset $\mathcal{D}$ comprising multiple samples $(x, y_0, y_1, i)$, where $y_0$ and $y_1$ are two completions under a given instruction $x$, and $i \in \{0, 1\}$ indicates that $y_i$ is the preferred completion out of the pair of $y_0$ and $y_1$. Motivated by the Bradley-Terry model (Bradley & Terry, 1952) on the formulation of human preference distrition $p^*(y_i \succ y_{1-i}|x)$, DPO aims to optimize the model $\theta$ starting from $\theta_0$ via

$$\min_\theta \mathbb{E}_{(x,y_0,y_1,i)\sim\mathcal{D}} \left[ \mathcal{L}_{DPO}\left(\theta|x,y_0,y_1,i\right) \stackrel{\triangle}{=} -\log\sigma\left(\beta\log\frac{\pi_\theta(y_i|x)}{\pi_{\theta_0}(y_i|x)} - \beta\log\frac{\pi_\theta(y_{1-i}|x)}{\pi_{\theta_0}(y_{1-i}|x)}\right)\right]. \quad (2)$$

## 4 FEDBIS: A VANILLA AND FEASIBLE FRAMEWORK FOR ACHIEVING FEDERATED RLHF

We aim to fine-tune an LLM using clients' preference data, enabling it to generate reader-friendly responses. Since preference data contain sensitive personal information, some clients may be hesitant to share this information due to privacy concerns. Recently, companies have developed on-device pretrained language models (e.g., Phi-3 (Abdin et al., 2024) and Qwen (Bai et al., 2023)), with the latest iPhone release integrating this technology (Gunter et al., 2024). This on-device feature allows clients to ask private and sensitive questions directly on their smartphones, ensuring that even the server (i.e., LLM owner) cannot access the input prompts (Wu et al., 2024a).

The proposed FedBis provides a simple yet effective solution through a three-step process to enable model fine-tuning with clients' preference data while preserving clients' privacy, as depicted in Figure 2: In the first step, each client builds their own *preference dataset*, which is originated from the daily queries (a.k.a. prompts or instructions) to a pretrained language model (LM), the model generated a pair of responses to each query, and the client chooses the preferred one. After the construction of the preference dataset, each client independently trains a *binary selector*, and the server aggregates these selectors into a global one and broadcasts it to the clients. This communication process repeats for a total of $R$ times, and the clients keep the preference dataset unchanged during the training. In contrast to traditional RLHF methods that train a regression-based reward model, this step employs a binary selector to alleviate computational constraints on edge devices (Wu et al., 2023b; 2024b; Wang et al., 2023b; 2024a; Li et al., 2024). Additionally, the selector enables comprehensive pairwise comparisons between completions, thereby facilitating more efficient training (Liu et al., 2024b; Dubois et al., 2024b). Afterward, we utilize the well-trained binary selector to enhance the performance of LLM. Specifically, we assume the server holds a set of instructions, together with pairwise responses generated by an LLM. Then, we build a preference dataset with the help of the binary selector and boost the LLM by means of DPO (Rafailov et al., 2023). This section highlights the key steps of the proposed FedBis, while Appendix A offers a detailed description.

**Local Training in FL.** Suppose client $m \in [M]$ holds a set of pairwise data with the size of $n_m$, i.e., $\hat{\mathcal{D}}_m = \{(x_i, y_{i,w}, y_{i,l})\}_{i \in [n_m]}$, where $x_i$ is the prompt, $y_{i,w}$ is the preferred completion out of the pair of $y_{i,w}$ and $y_{i,l}$. We reorganize these data and build a preference dataset $\mathcal{D}_m$ to be $\{(x_i, y_{i,w}, y_{i,l}, 0), (x_i, y_{i,l}, y_{i,w}, 1)|(x_i, y_{i,w}, y_{i,l}) \in \hat{\mathcal{D}}_m\}$ for training, in which each contains

the prompt, a pair of completions and preference selection. Apparently, this dataset eliminates the position effects, and we can train the selector as a classification task. Therefore, we utilize cross-entropy (CE) loss $\ell_{CE}$ to optimize the selector and formulate the expected loss as

$$F_m(\phi) = \mathbb{E}_{(x,y_0,y_1,i) \sim \mathcal{D}_m} \left[ \ell_{CE}(i|\phi; x, y_0, y_1) \right]. \tag{3}$$

**Generating synthetic preference data for DPO.**    Suppose the server holds a set of instructions $\hat{\mathcal{D}}$. With the LLM $\theta_0$, we can generate multiple completions for an instruction $x \in \hat{\mathcal{D}}$, resulting in a set of $n$ completions $(y_0, \ldots, y_{n-1}) \sim \pi_{\theta_0}(y|x)$. For each instruction, we can form a total of $\binom{n}{2}$ pairs of completions. We then use the binary selector $\phi_R$ to choose the optimal completion for each pair $(y_j, y_l)$ where $0 \le j < l \le n-1$. The pair is labeled with $i = 0$ if the first logit output is greater than the second, i.e., $\pi_{\phi_R}(0|x, y_j, y_l) > \pi_{\phi_R}(1|x, y_j, y_l)$, or $i = 1$ otherwise. This process builds the preference dataset $\mathcal{D}_{gen}$.

**Limitations.**    While the proposed FedBis effectively achieves RLHF in FL with low computational costs, there are two key limitations that remain unsolved. The first limitation is *preference heterogeneity*. The proposed work effectively aggregates client preferences via clients' preference models, but it does not address the preference gap across clients when there is a huge difference in client preferences and prompts. As a result, the clients optimize their local models for their own data instead of a global objective. This leads to global aggregation diverging from the global optimum, which lengthens the training time to obtain the desired performance (Karimireddy et al., 2020; Wu et al., 2023a). Another limitation is *reward hacking*. The proposed FedBis relies on training a single selector and using it to fine-tune the LLM. This procedure introduces an adversarial dynamic where the model "cheats" the selector into favoring certain responses without genuinely improving. Eventually, as the LLM is trained across more iterations, its performance degrades significantly, making the approach inefficient and unsustainable. In the coming section, we propose a new algorithm that is able to address both limitations while maintaining low computational costs.

## 5   FEDBISCUIT: FEDBIS WITH CLUSTER-WISE AGGREGATION

In this section, we aim to address the issues of preference heterogeneity and reward hacking in FedBis. To mitigate reward hacking, Eisenstein et al. (2023) and Coste et al. (2024) propose a promising method that trains multiple reward models simultaneously. Aggregating outputs from several models can lead to a more robust reward estimate. Additionally, recognizing that some clients may share similar preferences, we utilize clustered FL (Sattler et al., 2020; Ghosh et al., 2020; Ma et al., 2023) to group clients with common preferences for joint selector training. These two strategies complement each other, motivating us to combine them into a novel algorithm, FedBiscuit, which addresses both reward hacking and preference heterogeneity.

However, integrating these approaches is non-trivial, particularly when using existing clustered FL algorithms. Current algorithms predetermine the number of models and partition clients into groups, with each group training its own model. This one-to-one mapping assumes a fixed number of groups, but in practice, predefining the number of groups is challenging, and some models may end up untrained if no clients are assigned to them. If these untrained selectors are excluded, it could reduce the complexity of reward hacking, making it easier for the LLM to "cheat" the selector by favoring certain responses without genuine improvement. On the other hand, including untrained selectors could misalign the LLM, leading to incorrect alignments. Therefore, there is a need for a sustainable algorithm that ensures every selector contributes meaningfully to resisting reward hacking.

**Problem Formulation.**    In this work, we consider training multiple binary selectors $\phi_{[U]}$, which independently decide on a better completion out of a pair. It is noted that $U$ should be an odd number because this guarantees one completion is preferred by more selectors. Moreover, to ensure that all selectors are trained without bias towards a small specific group, we mandate that these selectors be trained using evenly disjoint clusters of clients. Additionally, a client's preference should align more closely with those within the same cluster than with those in different clusters. To this end, we can formulate the following objective:

$$\min_{\phi_{[U]} \in \mathbb{R}^{U \times d}} \quad F(\phi_{[U]}) \triangleq \sum_{m \in [M]} p_m \left( \min_{u \in [U]} F_m(\phi_u) \right) \tag{4}$$

$$s.t. \quad \max\{|M_u|\}_{u \in [U]} - \min\{|M_u|\}_{u \in [U]} \le 1, \tag{5}$$

---

**Algorithm 1** FedBiscuit

---

**Input:** local learning rate $\eta_l$, global learning rate $\eta_s$, local updates $K$, warm-up rounds $T$ for each binary selector, total communication rounds $R$, client regrouping interval $\tau$, pretrained LLM $\tilde{\phi}$.
**Require:** OPTIM$(m, \phi, K)$ fine-tunes model $\phi$ with the data of a client $m \in [M]$ for $K$ iterations and returns an optimized model.
**Require:** CG$(\phi_{[U]})$ assigns each client $m \in [M]$ to train one of the models $\phi_{[U]}$ and returns a list $\{U_m\}_{m \in [M]}$ indicating that a client $m$ should train the model $\phi_{U_m}$.

▷ Warm-up
1: **for each** $u \in [U]$ **do**
2:     Initialize the binary selector $\phi_{u,0} = \tilde{\phi}$
3:     **for** $t = 0, 1, \ldots, T-1$ **do**
4:         Sample clients $\mathcal{A} \subseteq [M]$
5:         Send $\phi_{u,t}$ to clients $m \in \mathcal{A}$
6:         **for** $m \in \mathcal{A}$ **in parallel do**
7:             $\phi_{u,t,K}^m = $ OPTIM$(m, \phi_{u,t}, K)$
8:             Send $\phi_{u,t,K}^m$ to the server
9:         **end for**
10:         $\phi_{u,t+1} = \frac{M}{A} \sum_{m \in \mathcal{A}} \phi_{u,t,K}^m$
11:     **end for**
12:     $\phi_u = \phi_{u,T}$
13: **end for**

▷ Clustered FL Training
14: Initialize $\phi_{u,0} = \phi_u$ for each $u \in [U]$
15: **for** $r = 0, 1, \ldots, R-1$ **do**
16:     **if** $r \% \tau == 0$ **then**
17:         $\{U_m\}_{m \in [M]} = $ CG$(\phi_{[U],r})$
18:     **end if**
19:     Sample clients $\mathcal{A} \subseteq [M]$
20:     Send $\phi_{U_m,r}$ to clients $m \in \mathcal{A}$
21:     **for** $m \in \mathcal{A}$ **in parallel do**
22:         $\phi_{U_m,r,K}^m = $ OPTIM$(m, \phi_{U_m,r}, K)$
23:         Send $\phi_{U_m,r,K}^m$ to the server
24:     **end for**
25:     Calculate each $\phi_{[U],r+1}$ via Equation (6)
26: **end for**

---

where the function $F_m$ follows the same definition of Equation (3). $\phi_u$ indicates the $u$-th binary selector, and $M_u$ means a set of clients using the $u$-th selector. By definition, $\cup_{u \in [U]} M_u = [M]$, and $\cap_{u \in [U]} M_u = \emptyset$. Next we explore how the proposed FedBiscuit optimizes Equation (4) under the constraint of Equation (5).

## 5.1 ALGORITHM DESIGN

Section 4 mentions that a client $m \in [M]$ holds a preference dataset $\mathcal{D}_m$. Before the model training, client $m$ splits her dataset into two disjoint sets, namely, a training set $\mathcal{D}_{m,train}$ and a validation set $\mathcal{D}_{m,val}$, where $|\mathcal{D}_{m,train}| >> |\mathcal{D}_{m,val}|$.

The proposed FedBiscuit consists of two phases: 1) We train each selector for a couple of rounds so that all $U$ selectors have basic capacities in selecting the preferred completion, and 2) we divide the clients into disjoint clusters of size $U$ and train each binary selector with a specific cluster. A full pseudocode implementation is provided in Algorithm 1.

**Phase 1: Warm-up.** In the beginning, we initialize each binary selector $\phi_u (u \in [U])$ with an identical pretrained LLM $\tilde{\phi}$. Subsequently, starting from $u = 0$, we train each selector $\phi_u$ for $T$ consecutive communication rounds following the steps of FedBis: In each round, the server samples a subset of clients $\mathcal{A}$ and broadcasts the selector $\phi_u$ to them. Each client $m \in \mathcal{A}$ then locally trains the selector for $K$ iterations using the dataset $\mathcal{D}_{m,train}$. At the end of the communication round, the server aggregates and updates the selector $\phi_u$. After completing the training of $\phi_u$, the server initiates the training of a new selector $\phi_{u+1}$ by repeating the above steps until all selectors are trained.

The selectors are trained with different data distributions because the clients participating in each training round are randomly selected. Consequently, all the selectors $\phi_{[U]}$ have distinct model parameters, leading to varied performance in terms of final logit output when given an instruction and a pair of completions.

**Phase 2: Clustered FL Training.** After the first phase, we obtain $U$ different selectors, denoted by $\phi_{[U],0}$. Unlike FedBis, this phase includes an additional step called *client grouping*, which partitions the clients into multiple disjoint clusters based on their preferences. In each communication round $r \in [R]$, FedBiscuit optimizes all the selectors $\phi_{[U],r}$ using the following four steps:

*Step 2.1: Client Grouping* CG$(\phi_{[U],r})$. This step is executed every $\tau$ communication rounds, i.e., when $r$ can be divided by $\tau$, or $\tau | r$. During this step, the server broadcasts all selectors $\phi_{[U],r}$ to all clients $[M]$. Then, a client $m$ calculates the averaged loss for each selector $\phi_{u,r}$ using local validation set via $\frac{1}{|\mathcal{D}_{m,val}|} \sum_{(x,y_0,y_1,i) \sim \mathcal{D}_{m,val}} [\ell_{CE}(i | \phi_{u,r}; x, y_0, y_1)]$. The server thereby collects

all these losses and adopts a greedy clustering approach (Sattler et al., 2020; Ma et al., 2023) to assign each client to the selector where they achieve the minimum loss. However, an obvious deficiency is an imbalance where some selectors are chosen by many clients and others by few. It is noted that the selectors trained with more clients achieve remarkable performance, while some may be overfitted to a specific group of clients. Therefore, the greedy clustering approach negatively impacts the overall performance when building a global preference dataset. To tackle the limitation, we propose to balance the clusters using the following steps repeatedly until the clients are evenly distributed: (i) Choose the cluster selected by the most clients, and (ii) If the cluster can accommodate $n$ clients, cap the cluster at $n$ clients and reassign the rest to other clusters where they achieve suboptimal loss. Finally, we obtain balanced and disjoint clusters. Let a client $m$ train the $U_m$-th selector $\phi_{U_m}$ for the next $\tau$ rounds. After this client grouping step, FedBiscuit proceeds to the following three steps.

*Step 2.2: Model Broadcast.* Similar to FedBis, the server samples $A$ clients from all clients $[M]$, denoted by $\mathcal{A}$. For each selected client $m \in \mathcal{A}$, the server transmits the selector $\phi_{U_m,r}$. This process can be characterized by defining $\mathcal{A}_u$ as the group of clients chosen to train the selector $\phi_u$. This ensures that $\cup_{u \in [U]} \mathcal{A}_u = \mathcal{A}$ and $\cap_{u \in [U]} \mathcal{A}_u = \emptyset$.

*Step 2.3: Local Training.* The client $m \in \mathcal{A}$ receives a binary selector $\phi_{U_m,r}$ from the server and trains the selector for $K$ iterations via $\phi_{U_m,r,k+1}^m = \phi_{U_m,r,k}^m - \eta \nabla F_m \left( \phi_{U_m,r,k}^m \right), k \in [K]$. Finally, let the updated local selector be $\phi_{U_m,r,K}^m$, and the client pushes it to the server.

*Step 2.4: Global Aggregation.* The server collects updated selectors from all participants $\mathcal{A}$. Since there are several binary selectors, the server updates each selector with a designated group of clients. For instance, the aggregation rule for the selector $u \in [U]$ follows

$$\phi_{u,r+1} = \left( 1 - \sum_{m \in \mathcal{A}_u} p_m \right) \phi_{u,r} + \sum_{m \in \mathcal{A}_u} p_m \phi_{u,r,K}^m. \tag{6}$$

It is noted that performance degradation occurs when a model is trained by clients with time-varying sizes in FedAvg (Gu et al., 2021; Wang & Ji, 2023). In other words, the weighted average aggregation strategy is no longer suitable for multi-selector aggregation due to the fluctuation in the number of clients training a specific selector in each communication round. Therefore, FedBiscuit adopts a new aggregation rule as formulated in Equation (6).

FedBiscuit finally produces a set of well-trained selectors $\phi_{[U],R}$ and the subsequent objective is to enhance LLM performance with the help of these selectors, as explored below.

**Reinforcement-learning Fine-tuning with Multiple Selectors.** We can leverage the methodology described in Section A.2, and one of the key steps involves constructing a preference dataset incorporating multiple selectors. For this, we employ a strategy of majority voting. Given an instruction $x \in \hat{\mathcal{D}}$ and a pair of generated completions $(y_0, y_1)$, we assume a selector $u \in [U]$ prefers $y_{i_u}$, where $i_u \in \{0, 1\}$. Therefore, the pair is assigned a label $i = \arg\max\{i_u\}_{u \in [U]}$, meaning that the completion $y_i$ is favored by most of the clients.

## 5.2 DISCUSSION: INTEGRATION WITH LORA

As all binary selectors are LLM, training them may consume significant communication and computation overheads. Besides, multiple LLMs lead to considerable storage burdens shouldered by the server. To reduce the costs, we adopt a parameter-efficient fine-tuning approach LoRA (Hu et al., 2021), where all binary selectors share the same base model while using different adapters.

In comparison with FedBis, FedBiscuit requires extra costs, i.e., $O(MU\lfloor R/\tau \rfloor \cdot C)$, where $C$ is the communication cost of a selector. This is because FedBiscuit involves client grouping periodically, unilaterally transferring all selectors from the server to the clients. Despite the extra costs, extensive experiments demonstrate non-trivial improvement by comparing FedBiscuit with FedBis.

## 6 FEDERATED HUMAN PREFERENCE BENCHMARK

In this section, we describe the preparation of federated human preference datasets, while the next section presents the experimental setup and quantitative analysis. We explore two open-ended text generation tasks, i.e., summarization and question-answering, based on publicly available datasets. Each task comprises two components: client preference data and unlabeled prompts. Below, we

outline the process of constructing a federated preference dataset. Detailed information on the datasets is provided in Table 5 in Appendix B.

**Summarization.** Stiennon et al. (2020) introduces a summarization dataset that consists of Reddit posts with human-written TL;DR (Völske et al., 2017). This dataset consists of two parts, one is a pretrained dataset, while the other is a dataset with human preference. As suggested by Ouyang et al. (2022), we ensure a post does not appear in both datasets. We assume the pretrained dataset is stored on the server side, and 60% of data are reserved for supervised fine-tuning (SFT). The remaining 40% are used for the RLHF process to improve LLM performance and generate human-preferred content. Since the human-preference dataset contains the worker ID, we partition the dataset based on the worker ID so that the dataset can be partitioned into 53 workers.

**Question-Answering (QA).** We reconstruct the public dataset SHP, which comprises numerous questions from Reddit posts and their corresponding user answers. The preference indicator is based on the number of likes an answer receives. Following the training of StreamSHP (Ethayarajh et al., 2022), we utilize the data with no more than 512 tokens. Given that the dataset spans 18 domains, we partition the dataset using a Dirichlet distribution with a parameter of 0.3, ensuring that no questions overlap between clients. In our experiment, we consider training the binary selector with 200 clients, which is a common setting when evaluating the performance of an FL algorithm (Jhunjhunwala et al., 2023). Figure 3 visualizes the data distribution on the selected clients. For the RLHF process, we incorporate 2.6K Reddit questions and 44.6K SafeRLHF prompts (Dai et al., 2023).

# 7 EXPERIMENTS

## 7.1 EXPERIMENTAL SETUP

**Model and computation environment.** We initialize the binary selector(s) using three pretrained base models, i.e., Qwen-2-0.5B (Bai et al., 2023), Gemma-2B (Team et al., 2024), and LLaMA-2-7B (Touvron et al., 2023), configuring the final layer to produce binary outputs "A" and "B" only. We adopt Gemma-2B and LLaMA-2-7B models for the summarization and QA tasks, where both models are fine-tuned on the Alpaca dataset (Taori et al., 2023). Our implementation is built upon FederatedScope (Xie et al., 2023; Kuang et al., 2023). The experiments are conducted on machines with one Nvidia A100 GPU card, Intel Xeon Platinum 8369B CPUs, and 256GB RAM.

**Baselines.** Since no prior work systematically enables RLHF in FL, we propose the following baselines, which extend previous studies to fit the FL and RLHF objectives. In each case, we directly optimize the pretrained model.

- **FedAvg:** Given that preference data on clients is pairwise, each client trains its local model to improve completions based on specific instructions. To minimize training costs, we employ LoRA for training and aggregation, following the approach in Sun et al. (2024).
- **FedDPO:** Clients train their local models using the DPO loss (as defined in Equation 2), and the server aggregates these local models into a global model using a weighted average. This method, incorporated in the OpenFedLLM benchmark (Ye et al., 2024), requires substantial computational resources and results in long local training wall-clock times.

**Evaluation.** We evaluate summarization and QA tasks using different datasets and methodologies:

- **Summarization task:** We use a test dataset consisting of 6,553 samples, all sourced from the TL;DR dataset and excluded from the training data. The model is tasked with generating summaries for each sample. Since human-labeled summaries are available, we measure the win rate of the model-generated summaries against the human ones using the Auto-J model (Li et al., 2023a).
- **QA task:** For question answering, we use AlpacaEval 2.0 (Li et al., 2023b; Dubois et al., 2024a;b) to assess model performance on 805 instructions. The model's responses are compared with those of GPT-4, and the win rate is calculated based on evaluations conducted by GPT-4-turbo-20240409.

**Implementation.** In our experiments, we train the binary selector for 500 communication rounds. In each round, we sample 5 clients for the summarization task and 10 for the QA task, and the selected clients fine-tune the binary selector locally for 30 iterations. As for FedBiscuit, the warm-up phase takes 50 communication rounds for each adapter, which is counted as part of 500 communication rounds. After the training of binary selectors, we fine-tune the LLM for three epochs, and we store the checkpoint when finishing one epoch of training. By default, the evaluation result reports the best-saved checkpoints. Due to the limited space, the details related to the hyperparameters are deferred to Appendix B.

| Binary Selector | Methods | Gemma-2B Win Rate (# Wins / # Ties) | LLaMA-2-7B Win Rate (# Wins / # Ties) |
|---|---|---|---|
| NA | Raw Model | 67.27% (4408 / 28) | 76.79% (5032 / 31) |
| | FedAvg | 28.66% (1878 / 10) | 28.23% (1850 / 10) |
| | FedDPO | 49.03% (3213 / 23) | 77.02% (5047 / 30) |
| Qwen-2 (0.5B) | FedBis | **86.69% (5681 / 26)** | 89.63% (5874 / 38) |
| | FedBiscuit ($U = 3$) | 78.45% (5141 / 29) | 83.29% (5458 / 48) |
| | FedBiscuit ($U = 5$) | 73.97% (4847 / 22) | 82.04% (5376 / 37) |
| Gemma (2B) | FedBis | 77.34% (5068 / 39) | 86.22% (5650 / 39) |
| | FedBiscuit ($U = 3$) | 83.81% (5492 / 43) | 91.32% (5984 / 28) |
| | FedBiscuit ($U = 5$) | 83.12% (5447 / 33) | 89.93% (5893 / 31) |
| LLaMA-2 (7B) | FedBis | 82.56% (5410 / 28) | **91.87% (6020 / 40)** |
| | FedBiscuit ($U = 3$) | 81.90% (5367 / 41) | 90.84% (5951 / 37) |
| | FedBiscuit ($U = 5$) | 79.31% (5197 / 41) | 90.54% (5933 / 47) |

Table 1: Performance under summarization task. **Bold** means the best result under the pretrained model; Underline means the best result under a binary selector.

## 7.2 QUANTITATIVE EVALUATION ON SUMMARIZATION TASK

Table 1 presents a comparative analysis of human-written and model-generated summaries, where the win rate indicates the likelihood that a generated summary surpasses its human counterpart, evaluated using the Auto-J metric. It is evident that both our proposed baseline and algorithm significantly outperform the raw model and other baselines. For both base LLMs – Gemma-2B and LLaMA-2-7B – all our methods demonstrate a performance improvement of at least 6% over the raw model, underscoring the effectiveness of our approach irrespective of the binary selector used. Notably, all baseline approaches exhibit a substantial decline in performance compared to the raw model, with a decrease of at least 20% in the win rate measurement.

We offer a plausible explanation for the performance drop of FedDPO, thereby highlighting the effectiveness of FedBis and FedBiscuit. Our proposed methods also incorporate DPO, differing significantly from FedDPO in terms of data distribution. While FedDPO relies directly on client preference datasets, our methods create a set of preference data comprising three components: unlabeled prompts, a pair of responses generated by the base model for each prompt, and preference selections simulated by the aggregated binary selector. Consequently, our generated responses align more closely with model outputs, facilitating easier guidance for the model to refine its responses into user-acceptable expressions. In contrast, FedDPO may confuse the model since it lacks outputs similar to those in the client preference datasets, leaving the LLM unsure of how to enhance the generated responses. An implicit assumption of DPO is that preference data should closely resemble model outputs; however, our proposed method may not be bound by this assumption.

**Performance analysis on various binary selectors.** Table 1 presents the importance that a selector trained with a proper method can significantly enhance the performance of base models (i.e., Gemma-2B and LLaMA-2-7B). A powerful selector does not mean that it can significantly boost the base model performance after alignment. For example, the smallest binary selector, Qwen-2, achieves a win rate of 86.69% under the Gemma-2B model, performing much better than the other two types. Different training methods on different binary selectors may lead to different effects. For instance, training Qwen-2 and LLaMA-2 with FedBis is always better than that of FedBiscuit, while training the Gemma selector with FedBiscuit ($U = 3$) would achieve the best performance.

## 7.3 QUANTITATIVE EVALUATION ON QA TASK

Table 2 shows that our methods, FedBis and FedBiscuit, consistently outperform the baseline approaches in terms of length-controlled win rate for both the Gemma-based model and the LLaMA-2 model. The baseline FedAvg performs worse than the Raw Model, with FedAvg showing a win rate of only 1.88% for the Gemma-based model and 3.93% for the LLaMA-2 model, compared to the Raw Model's 2.40% and 4.80%, respectively. FedDPO offers a slight improvement over the Raw Model, achieving a win rate of 3.28% for the Gemma-based model and 4.98% for the LLaMA-2 model, but it remains inferior compared to our methods. Specifically, FedBis achieves win rates of up to 4.58% for the Gemma-based model and 5.25% for the LLaMA-2 model, while FedBiscuit ($U = 3$) reaches

| Unlabeled Prompts | Methods | Gemma-2B LC Win-rate (%) | LLaMA-2-7B LC Win-rate (%) |
|---|---|---|---|
| NA | Raw Model | $2.40 \pm 0.16$ | $4.80 \pm 0.41$ |
| | FedAvg | $1.88 \pm 0.12$ | $3.93 \pm 0.23$ |
| | FedDPO | $3.28 \pm 0.23$ | $4.98 \pm 0.30$ |
| Reddit Posts (2 completions) | FedBis | $3.85 \pm 0.25$ | $5.06 \pm 0.30$ |
| | FedBiscuit ($U = 3$) | $3.70 \pm 0.24$ | $5.04 \pm 0.31$ |
| | FedBiscuit ($U = 5$) | $\underline{3.98 \pm 0.24}$ | $\underline{5.42 \pm 0.32}$ |
| Reddit Posts (4 completions) | FedBis | $3.66 \pm 0.25$ | $5.08 \pm 0.31$ |
| | FedBiscuit ($U = 3$) | $\underline{4.14 \pm 0.27}$ | $\underline{5.34 \pm 0.32}$ |
| | FedBiscuit ($U = 5$) | $3.80 \pm 0.27$ | $4.81 \pm 0.28$ |
| SafeRLHF | FedBis | $\mathbf{4.58 \pm 0.30}$ | $5.25 \pm 0.32$ |
| | FedBiscuit ($U = 3$) | $4.03 \pm 0.28$ | $\mathbf{5.63 \pm 0.34}$ |
| | FedBiscuit ($U = 5$) | $3.85 \pm 0.27$ | $5.42 \pm 0.35$ |

Table 2: Performance under QA task using AlpacaEval 2.0. "LC" in the table means "length-control." FedBis and FedBiscuit adopt the reward model of Gemma-2B fine-tuned on SHP dataset. **Bold** highlights the best result under each column, while Underline visualizes the best result under different sources of unlabeled prompts.

up to 4.14% and 5.63%, respectively. These results confirm that our methods provide significant improvements over the baselines, enhancing the length-controlled win rates for both models.

**Performance analysis on various unlabeled prompts.** Table 2 also shows that the use of different unlabeled prompts, such as Reddit posts and SafeRLHF, significantly affects the performance of our methods. When using Reddit prompts, generating 2 or 4 completions leads to variations in win rates. For example, with two completions, FedBiscuit ($U = 5$) achieves 3.98% for the Gemma-based model and 5.42% for the LLaMA-2 model. However, increasing to four completions slightly shifts the best performance for the Gemma model to FedBiscuit ($U = 3$) with 4.14%, while the win rate for the LLaMA-2 model slightly drops to 5.34%. On the other hand, SafeRLHF prompts consistently yield the best results overall, with FedBis achieving 4.58% for the Gemma model and FedBiscuit ($U = 3$) reaching 5.63% for the LLaMA-2 model. These findings demonstrate that SafeRLHF is the most effective prompt source, outperforming Reddit-based prompts.

## 7.4 ABLATION STUDY

Considering both tasks, the results reveal that the performance of FedBis and FedBiscuit varies across datasets. In the summarization task (Table 1), FedBis outperforms FedBiscuit across most binary selectors, achieving the highest win rates for both the Gemma-2B and LLaMA-2-7B models. For instance, under the Qwen-2 selector, FedBis reaches 86.73% for Gemma-2B and 89.67% for LLaMA-2-7B, while FedBiscuit ($U = 3$ and $U = 5$) are lower than those values. This is because preference heterogeneity is not critical in the summarization task. As described in Stiennon et al. (2020), the dataset is collected from a group of labelers who have a meeting from time to time to ensure they reach a consensus. However, in the QA task (Table 2), FedBiscuit proves to be the better method, particularly for the LLaMA-2-based model, where FedBiscuit ($U = 3$) achieves the highest win rate at 5.63% under SafeRLHF prompts, outperforming FedBis and other configurations. Although FedBis shows strength in some QA scenarios, such as the Gemma model with a 4.58% win rate, FedBiscuit demonstrates superior overall performance in the QA task. Therefore, FedBis is more effective in the summarization task, while FedBiscuit excels in the QA task.

## 8 CONCLUSION

In this work, we explore a feasible framework to employ a federated learning technique to enable large-scale user preference collection for RLHF without jeopardizing user privacy. Specifically, we train a binary selector across different clients using their local preference datasets, and then use the well-trained selector to align an LLM with human preferences. We propose two approaches to enable selector training: FedBis and FedBiscuit. FedBis provides a framework to train a single selector, while FedBiscuit ensembles multiple selectors to address preference heterogeneity and reward hacking. We conduct empirical studies with the proposed federated human preference datasets to validate our statements and demonstrate the superiority of FedBis and FedBiscuit when aligning Gemma-2B and LLaMA-2-7B with human preference.

## ETHICS STATEMENTS

This paper investigates clients' preferences using a publicly available dataset, ensuring that all data sources are appropriately cited to maintain academic integrity and transparency. By leveraging this public dataset, we avoid using private or sensitive client data, thus upholding ethical standards in data usage and research practices. Furthermore, this work prioritizes the protection of clients' privacy and strictly avoids any disclosure of local data. When clients utilize their own data to fine-tune the model, robust privacy measures are in place to ensure that no other clients can access or infer any information related to their data. This approach not only safeguards individual privacy but also fosters trust and security in the application of the model.

## ACKNOWLEDGEMENTS

The authors would like to thank the anonymous reviewers for their constructive comments. This work is supported in part by the US National Science Foundation under grants NSF CNS-2154059, NSF IIS-2141037, and NSF IIS-2226108. Any opinions, findings, and conclusions or recommendations expressed in this material are those of the author(s) and do not necessarily reflect the views of the National Science Foundation.

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

## A  A DETAILED IMPLEMENTATION OF FEDBIS

The objective of RLHF is to align a pretrained language model with human preferences. RLHF comprises two phases: (i) preference modeling and (ii) reinforcement-learning fine-tuning. The first phase aims to develop a model that simulates human preferences to select the superior options from numerous pairwise completions. Subsequently, the second phase enhances the language model's performance by creating a preference dataset, enabling the model to generate responses preferred by humans. The following describes the proposed FedBis that achieves RLHF in FL.

### A.1  PREFERENCE MODELING

We consider a practical and efficient FL scenario where not all clients but only a sampled subsets of clients participate in each communication round (Yang et al., 2020; Wu et al., 2023a; He et al., 2023; Fang et al., 2024). Before the commencement of FL training, we initialize the binary selector with a pretrained LLM such as LLaMA-2 (Touvron et al., 2023), and set the hyperparameters. An FL algorithm requires multiple communication rounds and consists of three phases in each round, i.e., *model broadcast*, *local training*, and *global aggregation*. Following this paradigm, we design FedBis and optimize the selector $\phi$, i.e., in the communication round $r \in [R]$, as discussed as follows.

**Step 1: Model Broadcast.**  The server uniformly samples $A$ clients without replacement, denoted by $\mathcal{A}$. Let the selector be $\phi_r$ in the $r$-th communication round, and the server broadcasts it to the sampled clients.

**Step 2: Local Training.**  At this step, client $m \in \mathcal{A}$ optimizes the selector based on local preference data. First, the client initializes the local selector $\phi_{r,0}^m$ with the global selector $\phi_r$ received from the server. Subsequently, the client trains the selector for $K$ iterations, where the update rule between consecutive iterations follows:

$$\phi_{r,k+1}^m = \phi_{r,k}^m - \eta \nabla F_m(\phi_{r,k}^m), k \in [K] \tag{7}$$

where the gradient $\nabla F_m(\phi_{r,k}^m)$ is approximated using a data batch sampled from the local preference dataset $\mathcal{D}_m$ and can incorporate optimizers such as AdamW (Loshchilov & Hutter, 2017). Finally, the client $m$ transmits the updated local selector $\phi_{r,K}^m$ back to the server.

**Step 3: Global Aggregation.**  After receiving the local selectors from the sampled clients $\mathcal{A}$, the server updates the global selector:

$$\phi_{r+1} = \frac{M}{A} \sum_{m \in \mathcal{A}} p_m \phi_{r,K}^m. \tag{8}$$

This aggregation method, based on Li et al. (2019) where the clients are uniformly sampled to train a global model, ensures consistency with Problem (1) in mathematical expectation. To prevent the server from inferring a client's preferences from their local model, a practical solution is secure aggregation, which ensures the server only accesses the aggregated results (Bonawitz et al., 2016; 2017; Kairouz et al., 2021; Chen et al., 2022; So et al., 2022; 2023; Rathee et al., 2023).

After $R$ communication rounds of training, FedBis outputs a binary selector $\phi_R$ that reflects the overall preferences of all clients. The selector can then be used to enhance the performance of the LLM, as discussed in the next section.

### A.2  REINFORCEMENT-LEARNING FINE-TUNING

The reinforcement-learning fine-tuning takes place on the server and includes two phases: 1) a preference dataset is created with a pretrained LLM $\theta_0$ and a well-trained selector $\phi_R$ from FedBis. 2) LLM is optimized according to the objective defined in Equation (2) with the generated dataset.

**Step 1: Preference Dataset Generation.**  Suppose the server holds a set of instructions $\hat{\mathcal{D}}$. With the LLM $\theta_0$, we can generate multiple completions for an instruction $x \in \hat{\mathcal{D}}$, resulting in a set of $n$ completions $(y_0, \ldots, y_{n-1}) \sim \pi_{\theta_0}(y|x)$. For each instruction, we can form a total of $\binom{n}{2}$ pairs of completions. We then use the binary selector $\phi_R$ to choose the optimal completion for each pair $(y_j, y_l)$ where $0 \leq j < l \leq n-1$. The pair is labeled with $i = 0$ if the first logit output is greater than the second, i.e., $\pi_{\phi_R}(0|x, y_j, y_l) > \pi_{\phi_R}(1|x, y_j, y_l)$, or $i = 1$ otherwise. This process builds the preference dataset $\mathcal{D}_{gen}$.

**Step 2: LLM Fine-tuning.** With the constructed preference dataset $\mathcal{D}_{gen}$, we evolve the LLM to align with clients' preferences. Specifically, in the $t$-th training round, where $t \in \{0, 1, \dots\}$, we sample a data batch $(x, y_0, y_1, i)$ from $\mathcal{D}_{gen}$, and update the LLM using the following rule:

$$\theta_{t+1} = \theta_t - \eta \nabla \mathcal{L}_{DPO}\left(\theta_t | x, y_0, y_1, i\right), \tag{9}$$

where $\eta$ is the learning rate. The gradient computation $\nabla \mathcal{L}_{DPO}$ is given by Rafailov et al. (2023). In a nutshell, we distill the binary selector's preferences into the LLM, allowing it to function as a binary selector itself implicitly.

# B  IMPLEMENTATION DETAILS AND HYPERPARAMETERS

In this section, we include various settings, such as the prompt and the hyperparameters.

## B.1  HYPERPARAMETER SETTINGS

In our work, we fine-tune all models using LoRA, which is consistently set to rank 8, $\alpha = 16$, and the dropout rate 0.05. For the generation, we apply these parameters:

- If it is required to generate multiple completions, then we set the temperature to 0.7. We set the maximum new tokens for 80 under the summarization task and 300 for QA tasks.

- If it is required to generate a single completion, then we adopt greedy search by setting the temperature to 0.0.

In the following part, we show the hyperparameter setting for different tasks:

|  | Selector Training | RLFT |
|---|---|---|
| Participation Rate | 5/53 | - |
| Local Iterations | 30 | - |
| Batch Size | 32 | 32 |
| Rounds | 500 | 5 epochs |
| Optimizer | AdamW | RMSprop |
| Hyperparameters | (0.9, 0.95) | - |
| Learning rate | $1e-5$ | $1e-6$ |

Table 3: Hyperparameter Settings for the Summarization Task

|  | Selector Training | RLFT |
|---|---|---|
| Participation Rate | 10/200 | - |
| Local Iterations | 30 | - |
| Batch Size | 32 | 32 |
| Rounds | 500 | 5 epochs |
| Optimizer | AdamW | RMSprop |
| Hyperparameters | (0.9, 0.95) | – |
| Learning rate | $1e-5$ | $1e-6$ |

Table 4: Hyperparameter Settings for the QA Task

## B.2  DATASET DETAILS

In Section 6, we discuss how to partition the dataset for two tasks, namely, summarization and QA. Table 5 comprehensively presents the dataset details of both tasks, while Figure 3 visualizes the data distribution of the selected clients.

| Task | Preference Dataset | # preference samples | # clients | Partition Rules | Max. | Min. | Std. | Unlabelled Dataset | # unlabelled prompts |
|---|---|---|---|---|---|---|---|---|---|
| Summarization | TL;DR comparison | 92858 | 53 | Worker ID | 12985 | 1 | 2284.33 | Open-AI TL;DR | 42782 |
| Question Answering | SHP | 260814 | 200 | Dirichlet(0.3) on categories | 4393 | 260 | 832.39 | SHP Test | 4293 |
|  |  |  |  |  |  |  |  | SafeRLHF | 44578 |

Table 5: Dataset details for federated human preference benchmark

**Special Setting for FedBiscuit.** For the above two tasks, we ensemble three binary selectors (i.e., LoRAs). In the warmup round, we train the selector for 50 rounds under an FL framework. FedBiscuit performs regrouping every 50 rounds in the summarization task, while regrouping every 100 rounds in the QA task.

## B.3  MORE EXPERIMENTS

**Comparison between two different model sizes.** Table 6 compares the performance between two Qwen-2 models as the selectors with different sizes (i.e., 0.5B and 1.5B) when using them to fine-tune a Gemma-2B model for the summarization task. These results show that while larger binary selectors (e.g., Qwen-2-1.5B) sometimes provide slight performance improvements, smaller selectors like Qwen-2-0.5B remain competitive, particularly when applying FedBis. These findings suggest that both model type and size influence the selector's effectiveness, and they need to be carefully balanced based on task requirements and resource constraints.

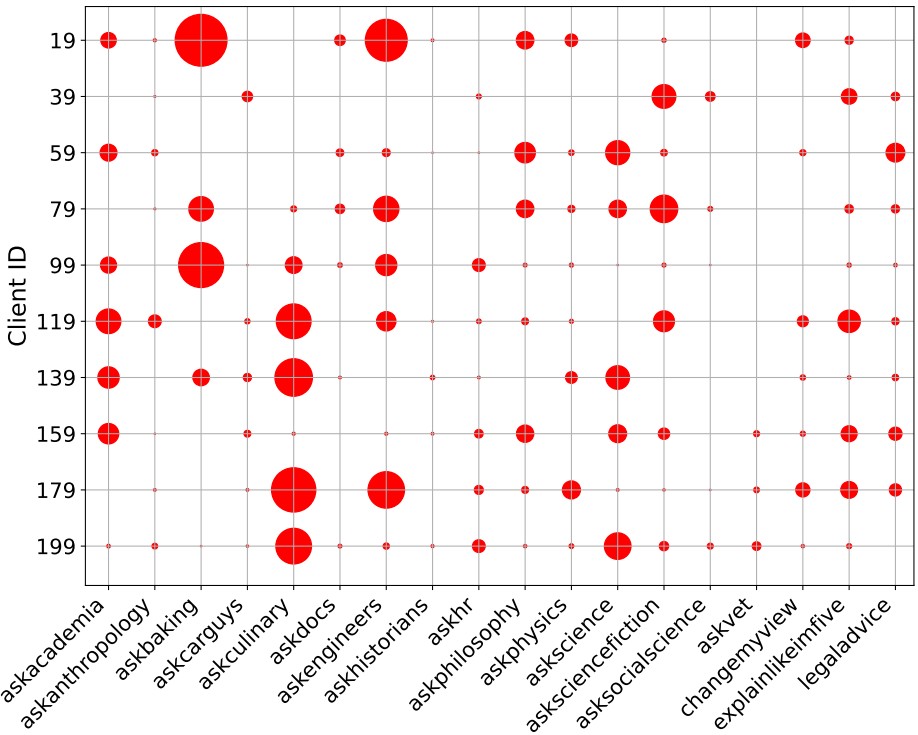

Figure 3: Data distribution across different question domains on the selected clients.

| Methods | Win Rate on Qwen-2 (0.5B) (# Wins / # Ties) | Win Rate on Qwen-2 (1.5B) (# Wins / # Ties) |
|---|---|---|
| FedBis | **86.69% (5681 / 26)** | **82.45% (5401 / 34)** |
| FedBiscuit ($U = 3$) | 78.45% (5141 / 29) | 80.18% (5254 / 29) |
| FedBiscuit ($U = 5$) | 73.97% (4847 / 22) | 76.22% (4995 / 43) |

Table 6: Performance using Gemma-2B to summarize a Reddit post (i.e., summarization task). **Bold** means the best result under the selector.

### B.4 INSTRUCTION TUNING PROMPT

The proposed work follows a previous study (Zhang et al., 2024) to fine-tune the model following a given prompt template. This is also known as instruction tuning. The prompt template is different between tasks and between the selector and base model. As a result, we provide the prompts in Figure 4 for detailed study.

## C SAMPLE DIALOGS

We provide sample dialogs for our methods, FedBiscuit and FedBis, compared against the baselines FedDPO, FedAvg, and the raw model, highlighting their performance in summarization and question-answering tasks. Our methods consistently deliver superior results, as demonstrated by the provided samples.

FedBiscuit generates precise and well-organized summaries, making it particularly effective in distilling complex information into clear and actionable insights. FedBis, while maintaining a similar level of clarity, offers more nuanced and in-depth responses, making it versatile for a wider range of queries. In contrast, the baseline models—FedDPO, FedAvg, and the raw model—tend to produce less coherent responses. FedDPO sometimes lacks structure, FedAvg oversimplifies the content, and the raw model can miss critical details, leading to less accurate results.

These samples provide clear evidence that FedBiscuit and FedBis offer better summarization and question-answering performance by consistently striking the right balance between conciseness, accuracy, and detail, proving to be more reliable and contextually appropriate than the baselines.

Prompts template of LLM generation for summarization task.

Below is an instruction that describes a task, paired with an input that provides further context. Write a response that appropriately completes the request.

### Instruction:
Summarize the following Reddit post in a paragraph of 50 words or less.

### Input:
SUBREDDIT: r/{subreddit}
TITLE: {title}
POST: {post}

### Response:

Prompt template when training a binary selector for summarization task.

Below is a forum post followed by two summaries. Pick a more precise and concise one that summarizes the most important points in the given forum post, without including unimportant or irrelevant details. State your choice with a single capital letter, i.e., "A" if SUMMARY A is better, "B" if SUMMARY B is better.

### SUBREDDIT: r/{subreddit}
### TITLE: {title}
### POST: {post}
### SUMMARY A: {output_A}
### SUMMARY B: {output_B}
### YOUR CHOICE:

Prompts template of LLM generation for QA task with an additional input.

Below is an instruction that describes a task, paired with an input that provides further context. Write a response that appropriately completes the request.

### Instruction:
{instruction}

### Input:
{input}

### Response:

Prompts template of LLM generation for QA task without an additional input.

Below is an instruction that describes a task. Write a response that appropriately completes the request.

### Instruction:
{instruction}

### Response:

Prompt template when training a binary selector for QA task with SHP dataset.

Below is a query followed by two responses. Pick a helpful response that is precise, concise, and casual. State your choice with a single capital letter, i.e., "A" if RESPONSE A is better, "B" if RESPONSE B is better.

### QUERY: {instruction}
### RESPONSE A: {output_A}
### RESPONSE B: {output_B}
### YOUR CHOICE:

Figure 4: Prompt Templates.

**User Input**

Give me a brief scenario of a persona that would search this information and find this content helpful: {NSW Residential Building Contract for Small Works over $20,000 Designed for residential building work which is not complex over $20,000. Suitable for smaller alteration and renovation projects. Small Works Contracts (Form 10A) Pricing structure Fixed price Value range Under $7,500 Use for Minor residential alterations and additions work Includes Succinct, plain-language conditions suitable for very small, low-risk projects in place of a quotation}

**FedBiscuit**

Imagine you are a homeowner who wants to make some changes to your home, but you don't want to spend a lot of money on a major renovation. You come across a small works contract that is designed for residential building work that is not complex and has a value range of under $7,500. This contract is suitable for smaller alteration and renovation projects. It is a fixed price contract, meaning the price is set and there are no hidden costs or additional fees.

The contract is written in plain language, making it easy to understand and use. It includes succinct conditions that are suitable for very small, low-risk projects. This means that if something goes wrong, you can easily resolve the issue and the contract will protect you.

Overall, this small works contract for residential building work is a great option for homeowners who want to make some changes to their home without spending a lot of money. It is a simple, straightforward solution that provides peace of mind and protection for your project.

**FedBis**

Imagine you are a homeowner who wants to make some changes to your home, but you don't want to spend a lot of money on a major renovation. You come across a small works contract that is designed for residential building work that is not complex and has a value range of under $7,500. This contract is suitable for smaller alteration and renovation projects. It is a fixed price contract, meaning the price is set and there are no hidden costs or additional fees.

The contract is written in plain language, making it easy to understand and use. It includes succinct conditions, which are suitable for very small, low-risk projects. The contract is intended for residential work, meaning it is not suitable for larger, more complex projects.

Overall, this small works contract for residential building work is a cost-effective and straightforward option for making small changes to your home without breaking the bank.

**FedDPO**

A persona that would find this information helpful is a homeowner who is planning to make some minor changes to their home, such as adding a new room or making some renovations. They are looking for a simple and straightforward contract that outlines the terms and conditions of the work, with a fixed price and a value range of under $7,500. They want to ensure that the contract is suitable for their small, low-risk project and that it is concise and easy to understand.

**FedAvg**

I'm not sure if this is helpful, but I'm a lawyer and I've been looking for a similar contract for a while now. I've found a few, but they're not quite what I'm looking for.

**Raw Model**

A persona that would search this information and find it helpful is a small residential building contractor who is looking to complete a small works project on their home. They are looking for a fixed price contract that is suitable for their project, which is not complex and will not exceed $20,000. They want a contract that is simple and easy to understand, with concise conditions that outline the scope of work and the expected timeline. They are looking for a contract that is suitable for small, low-risk projects, and they want to avoid the need for a quotation.

Figure 5: Gemma QA Sample.

**User Input**

Hi, I have some falafel, but no tahini to put on them. Can you give me a recipe for making tahini?

**FedBiscuit**

Sure, here is a recipe for making tahini at home:

Ingredients:
- 1 cup of sesame seeds
- 2 tablespoons of water
- 2 tablespoons of honey
- 2 tablespoons of lemon juice

Instructions:
1. Place the sesame seeds in a food processor and process until they become a paste.
2. Add the water, honey, and lemon juice and process until the mixture is smooth.
3. Taste and adjust the seasoning if necessary.
4. Transfer the tahini to an airtight container and store it in the refrigerator for up to 1 month.

**FedBis**

Sure, here is a recipe for making tahini at home:

Ingredients:
- 1 cup of sesame seeds
- 2 tablespoons of water
- 2 tablespoons of honey
- 2 tablespoons of lemon juice
- 1 teaspoon of garlic
- 1/2 teaspoon of salt

Instructions:
1. Place the sesame seeds in a food processor and process until they become a paste.
2. Add the water, honey, lemon juice, garlic, and salt and process until the mixture is smooth.
3. Taste and adjust the seasoning if necessary.
4. Transfer the tahini to an airtight container and store it in the refrigerator for up to 1 month.

**FedDPO**

You can make tahini at home by soaking a few tablespoons of sesame seeds in a small amount of water for 30 minutes, then grinding them in a food processor.
Add a tablespoon of the resulting paste to your falafel to give them a delicious, nutty flavor.

**FedAvg**

I've been making tahini with a blender.
I put a cup of sesame seeds in a blender, add a cup of water, and blend until smooth.
I then strain the mixture through a cheesecloth-lined strainer.
I've been using the tahini for a few weeks now and it's been great.

**Raw Model**

You can make tahini at home by soaking a few tablespoons of sesame seeds in a small amount of water for 30 minutes, then grinding them in a food processor.
Add a tablespoon of the resulting paste to your falafel to give them a nutty flavor.

Figure 6: LLaMA QA Sample.

**Reddit Post: What is your most memorable act of kindness and why?**

Random or otherwise.
I was walking to school one morning when I was about 15 when I heard a voice call out "Hello? Excuse me? Young man?".
I turned and saw a frail old man who must have been in his 90s standing in the doorway of his house.
I walked over and he asked if I could help him with something. I said I would help, so I followed him to a step-ladder in his kitchen.
He said that he needed to change the bulb but he couldn't get up the ladder.
After I changed the bulb, he said that he'd give me something for helping out.
He reached for a small tray of change on his kitchen table and picked out the only 50p coin in his tray of 1p and 2p coins.
I told him he didn't have to give me anything but he said that it was only right to reward helping out an old man.

**FedBiscuit**

A Reddit user recounts a heartwarming act of kindness where they helped an elderly man change a lightbulb in his kitchen.
The stranger gave them a 50p coin as a reward for their help.
The act of kindness was memorable because it brought a smile to the elderly man's face and showed that even in their old age, people can still do good deeds for others.

**FedBis**

A Reddit user recounts a heartwarming act of kindness where they helped an elderly man change a lightbulb in his kitchen. The stranger gave them a 50p coin as a reward for their help. The act of kindness was memorable because it brought a smile to the elderly man's face and showed that even in their old age, people can still do good deeds for others.

**FedDPO**

A Reddit user recounts a heartwarming story of a 90-year-old man who needed help changing a lightbulb in his kitchen.
The user, who was 15 at the time, heard the man's plea for help and went to his aid. The man gave the user

**FedAvg**

Old man in his 90s needed help changing a bulb. I helped him and he gave me a 50p coin.

**Raw Model**

A Reddit user recounts a heartwarming story of a stranger who helped an elderly man change a lightbulb in his kitchen.
The stranger, who was walking to school at 15, heard a voice call out for help and followed the old man to his house.
He changed the lightbulb and the old man gave him a 50p coin as a reward.
The stranger didn't have

Figure 7: Gemma Summarization Sample.

**Reddit Post: TIFU by locking myself out of my room.**

It all started with a cultural tour. Being active duty military and stationed overseas in Japan we were voluntold to partake in a tour of the local area. Fun and dandy right?

I was ecstatic about the free tour of the local area. Except when I woke up five minutes before the tour starts. Luckily the meeting area was a Starbucks just outside of my room.

Que the rushing, the frantic panic to dress up, and get down in time. I left my ID which is also conveniently my room key. Two more mistakes. (Straying away from the story, the military is all about being on time and integrity. Hence "excellence in all we do".)

Realizing I fucked up bad, I told the host that I had left my REQUIRED ID in my room, which is my room key locking myself out.

I couldn't go on the tour and now I'm stuck in the Starbucks waiting until one of my supervisors picks me up and lets me into my room. I'm expecting more than just a little scolding, paperwork even.

**FedBiscuit**

The poster was locked out of their room due to leaving their ID, which was also their room key, in their rush to make it on time for a cultural tour.

They are now stuck in a Starbucks waiting for someone to pick them up and let them into their room.

They expect to receive a scolding and possibly even paperwork for the mistake.

**FedBis**

While on a cultural tour in Japan, the poster locked themselves out of their room, leading to them being stuck in a Starbucks and expecting a scolding from their supervisors.

**FedDPO**

While on a cultural tour in Japan, the poster was locked out of their room due to leaving their ID and room key in the room. They are now stuck in a Starbucks waiting for a supervisor to pick them up and let them into their room. They expect to receive a scolding and possibly paperwork for the mistake.

**FedAvg**

I locked myself out of my room and now I'm stuck in Starbucks waiting for a supervisor to pick me up and let me into my room.

**Raw Model**

While on a cultural tour in Japan, the poster was locked out of their room due to leaving their ID and room key in the room.

They were stuck in a Starbucks until one of their supervisors picked them up and let them into their room.

They expect to receive a scolding and possibly paperwork for the mistake.

Figure 8: LLaMA Summarization Sample.

