# OpenReview forum: "Towards Federated RLHF with Aggregated Client Preference for LLMs"
_ICLR.cc/2025/Conference — ICLR 2025 Poster_

### Official Review · Reviewer_yFna · 2024-10-28

**Soundness:** 3
**Presentation:** 3
**Contribution:** 3
**Rating:** 6
**Confidence:** 2

**Summary:**

This paper proposes a federated RLHF method specially designed for large-scale preference collection, which encodes each client’s preferences into binary selectors and aggregates them to capture common preferences. An enhanced method is also introduced, which overcomes the challenges of preference heterogeneity and reward hacking by grouping similar clients and leveraging multiple binary selectors to improve LLM output quality. The authors also established a novel federated RLHF benchmark for evaluation. Experimental results show the superiority of the proposed model.

**Strengths:**

-The workload is impressive, every part is introduced in detail.

-The idea of training binary selectors to encode client’s preferences is novel.

-The two proposed methods are reasonable and both have many details in addressing the issues faced by traditional method.

**Weaknesses:**

-Computation overhead. This method involves additional training of binary selectors, which are pre-trained LLMs. Although the authors apply LoRA to reduce the computation cost, the time and computation overhead of fine-tuning LLM selectors are still not negligible, which limits its application scenario.

-The experiments need supplements: In the section of performance analysis on different binary selectors, the experiments applied three base model with both different model structure and parameter size. To make the results more convincing, the effect of model type and model size should be explored separately.

**Questions:**

-Is there any data available on the training time or memory footprint of the entire training process?

-Given that LoRA configuration also has an impact on performance, can the authors provide the detailed information about the LoRA configuration used in their experiments?

---

> ### Author Response · Authors · 2024-11-20
>
> > **[W1]** Computation overhead. This method involves additional training of binary selectors, which are pre-trained LLMs. Although the authors apply LoRA to reduce the computation cost, the time and computation overhead of fine-tuning LLM selectors are still not negligible, which limits its application scenario.
>
> Thanks for your question. The primary objective of federated RLHF is to enhance the content generation capabilities of LLMs. Training a very large model directly on clients' devices is indeed impractical due to resource constraints. To address this, we propose training a smaller, more sustainable binary selector model. While the binary selector is derived from a pre-trained LLM, it is significantly smaller and, therefore, trainable and maintainable by clients without prohibitive computational costs.
>
> To further clarify, as shown in Table 1, we demonstrate the effectiveness of training a small LLM, such as Qwen-0.5B, as a binary selector to enhance the performance of much larger base models like Gemma-2B and LLaMA-2-7B. Despite its smaller size, Qwen-0.5B can still provide substantial performance improvements. This approach significantly reduces the computational burden compared to directly fine-tuning a full-scale LLM.
>
> > **[W2]** The experiments need supplements: In the section of performance analysis on different binary selectors, the experiments applied three base model with both different model structure and parameter size. To make the results more convincing, the effect of model type and model size should be explored separately.
>
> Thanks for your suggestion. In this paper, we analyze various model types for binary selectors, as detailed in Section 7.2. It is worth noting that the models analyzed are at their smallest size, ensuring a fair comparison of model types.
>
> To further explore the impact of model size independently, we conducted additional experiments using two sizes of Qwen-2 models (0.5B and 1.5B) as binary selectors. Following the experimental setup described in Section 7.1, we evaluated these selectors on their ability to enhance open-ended text generation with Gemma-2B. The results, summarized below for the summarization task, provide insights into the effect of model size under each proposed method:
>
> |Win Rate (# Wins / # Ties)|Qwen-2-0.5B|Qwen-2-1.5B|
> |---|---|---|
> |FedBis|86.69% (5681 / 26)|82.45% (5401 / 34)
> |FedBiscuit ($U=3$)|78.45% (5141 / 29)|80.18% (5254 / 29)
> |FedBiscuit ($U=5$)|73.97% (4847 / 22)|76.22% (4995 / 43)
>
> From these results, we observe that while larger binary selectors (e.g., Qwen-2-1.5B) sometimes provide slight performance improvements, smaller selectors like Qwen-2-0.5B remain competitive, particularly when applying FedBis. These findings suggest that both model type and size influence the selector's effectiveness, and they need to be carefully balanced based on task requirements and resource constraints.
>
> > **[Q1]** Is there any data available on the training time or memory footprint of the entire training process?
>
> Thanks for your question. Below, we provide detailed measurements based on additional experiments conducted with a 1.5B Qwen-2 model.
>
> - **Memory footprint:** Under a batch size of 16, the peak GPU memory consumption is 56.7GB, 59.5GB, and 62.4GB when training the binary selector with FedBis, FedBiscuit ($U=3$), and FedBiscuit ($U=5$), respectively.
> - **Training Wall-clock Time:** For a 30-iteration local update, the wall-clock time is approximately 26 seconds. FedBiscuit requires client grouping every 50 rounds, adding a computation time of 30 seconds (for three adapters) and 52 seconds (for five adapters) to compute the losses if all clients compute the losses in parallel. It is worth noting that our federated learning experiments are conducted in a simulated environment, and therefore, the communication time between the server and clients is not included in these measurements. Moreover, our training is entirely on a Nvidia A100 GPU with a memory of 80GB.
>
>  > **[Q2]** Given that LoRA configuration also has an impact on performance, can the authors provide the detailed information about the LoRA configuration used in their experiments?
>
> Thanks for your question. As detailed in Appendix B.1, we fine-tune all models using LoRA with the following consistent settings: rank = 8, $\alpha = 16$, and a dropout rate of 0.05. These parameters were chosen to balance performance and computational efficiency across all experiments.

---

> > ### Author Response · Authors · 2024-11-24
> > **We look forward to your acknowledgement**
> >
> > Dear Reviewer yFna,
> >
> > Thank you for your time and insightful comments on our work. As we approach the end of the author-reviewer discussion period, we greatly value the opportunity to address your concerns.
> >
> > Could you kindly review our responses to see if they address your comments? We would highly appreciate it if you find our responses satisfactory and consider updating your rating. Feel free to reach out if you have any other questions or need further clarification.
> >
> > Best,
> >
> > Authors

---

> > ### Author Response · Authors · 2024-11-27
> > **We look forward to your follow-up comments**
> >
> > Dear Reviewer yFna,
> >
> > Thanks a lot for your commitment to reviewing our paper and providing insightful feedback. We hope this message finds you well.
> >
> > We would like to kindly remind you that we have less than 24 hours remaining to finalize our manuscript. Based on your valuable feedback, we have thoroughly revised our paper, and the updated content has been highlighted in red. Moreover, we have provided detailed responses to your comments. We are eager to hear your feedback to see if our revisions and responses have fully addressed your concerns, and we hope you might consider increasing your rating if the changes meet your expectations.
> >
> > If you have any further suggestions or additional concerns, please do not hesitate to let us know. We deeply value your comments and will try to address any remaining points.
> >
> > Thank you once again for your time and efforts.
> >
> > Best,
> >
> > Authors

---

> ### Author Response · Authors · 2024-12-03
> **Gentle Reminder for Discussion**
>
> Dear Reviewer yFna,
>
> We sincerely appreciate your time and efforts in reviewing our paper and providing valuable comments and suggestions. Based on your insightful feedback, we have revised our paper and addressed your concerns in our rebuttal. A summary of our responses is as follows:
>
> - **Computation overhead:** Compared to fine-tuning an LLM directly using human preference data, training a binary selector is significantly more lightweight, even though it inherits from a pretrained LLM. For example, our experiments demonstrate that the Qwen-2-0.5B selector provides strong performance enhancements to larger base models like Gemma-2B while ensuring computationally efficient. This approach reduces the training burden for federated RLHF and broadens its applicability to resource-constrained scenarios.
> - **Comparison of model sizes:** To address your suggestion, we conducted additional experiments comparing Qwen-2-0.5B and Qwen-2-1.5B selectors in the Gemma-2B generation task for summarization. Experimental results suggest that larger binary selectors can improve LLM generation performance compared to smaller ones.
> - **Runtime statistics:** Training a binary selector requires 56.7–62.4GB peak memory (depending on the method and models) and about 26 seconds for 30 local iterations.
> - **LoRA configuration details:** We used consistent LoRA settings across all experiments: rank = 8, $\alpha = 16$, and dropout rate = 0.05.
>
> For more details, please feel free to take a look at the above **Official Comment by Authors**.
>
> As the discussion will end in **8 hours**, we would like to kindly remind you to review our revisions and rebuttal and provide feedback on whether our revisions and rebuttal address your concerns. If you have any additional questions/concerns/suggestions, please do not hesitate to let us know. We will do our best to respond promptly and address them.
>
> Best,
>
> Authors of **Paper 13225 (Towards Federated RLHF with Aggregated Client Preference for LLMs)**

---

### Official Review · Reviewer_EwYt · 2024-11-02

**Soundness:** 3
**Presentation:** 3
**Contribution:** 3
**Rating:** 6
**Confidence:** 4

**Summary:**

This paper leverages federated learning into reinforcement learning with human feedback (RLHF) and proposes two federated-based RLHP methdos (FedBis and FedBiscui) to fine-tune large language models using user preference while preserving data privacy. The methods employ binary selectors to aggregate client preferences without requiring direct data sharing. FedBiscuit extends the functionality of FedBis by addressing preference heterogeneity and mitigating reward hacking. Experiments demonstrate that FedBis and FedBiscui perform better on Summarization task and QA task.

**Strengths:**

1、Generally, this paper is well-written with clear explanations of the methodologies and results.

2、The proposed FedBis and FedBiscui demonstrate good performance on several benchmarks, suggesting good-quality research and implementation.

3、This paper addresses the heterogeneity issue, an important issue in all fields of federated learning related research.

**Weaknesses:**

1、It would be better to include experiments that explore the sensitivity of the hyperparameters, particularly the number of clusters $U$, which is a crucial or even central parameter in the FedBiscuit method. However, the current results only present cases for $U=3$ and $U=5$, providing limited insights from these two configurations.

2、The authors should further explain why FedBiscuit with $U=3$ performs better in some cases while $U=5$ yields superior results in others. Since FedBiscuit addresses the heterogeneity issue, and as noted in existing federated learning literature, particularly in cluster-based federated learning approaches, the number of clusters is a key factor. Understanding the impact of different cluster sizes on the method’s effectiveness would be valuable.

3、The additional complexity introduced by clustering and the use of multiple selectors in FedBiscuit may still pose scalability challenges, particularly in real-world applications with thousands of clients. It would be helpful to provide more analysis on the scalability of the approach, especially under varying numbers of clients and communication costs. This would shed light on the method’s feasibility for large-scale deployment.

**Questions:**

Please see weaknesses.

---

> ### Author Response · Authors · 2024-11-20
> **Official Comment by Authors (1/2)**
>
> > **[W1]** It would be better to include experiments that explore the sensitivity of the hyperparameters, particularly the number of clusters $U$, which is a crucial or even central parameter in the FedBiscuit method. However, the current results only present cases for $U=3$ and $U=5$, providing limited insights from these two configurations.
>
> Thank you for your insightful comment. We agree that exploring the sensitivity of hyperparameters like the number of clusters $U$ is crucial for understanding FedBiscuit's behavior.
>
> First, $U$ is intentionally constrained to odd numbers to ensure that, for any pair of completions, one can be consistently preferred by a majority of selectors. Second, as shown in Table 1, FedBiscuit with $U=5$ performs worse than both FedBis and FedBiscuit with $U=3$. This can be attributed to the following: in Table 1’s experimental setup, where five clients are sampled per round, selectors often receive insufficient training when $U=5$. This results in inaccurate preference evaluations, which negatively affect the selection process of pairwise completions.
>
> Furthermore, based on the patterns observed, increasing $U$ beyond 5 would likely exacerbate these issues, leading to further performance degradation and higher computational overhead, as detailed in Section 5.2. Thus, expanding the binary selector size beyond this range would be both ineffective and inefficient.
>
> > **[W2]** The authors should further explain why FedBiscuit with $U=3$ performs better in some cases while $U=5$ yields superior results in others. Since FedBiscuit addresses the heterogeneity issue, and as noted in existing federated learning literature, particularly in cluster-based federated learning approaches, the number of clusters is a key factor. Understanding the impact of different cluster sizes on the method’s effectiveness would be valuable.
>
> Thanks for your question. In our experiments, $U=3$ generally outperforms $U=5$,  as indicated in the quantitative results. As discussed in our response to **[W1]**, larger $U$ values often lead to insufficient training for individual selectors, resulting in degraded performance. Additionally, larger $U$ values introduce substantial computational costs, making them impractical for real-world applications.
>
> However, using smaller values of $U$ is not without trade-offs. For instance, in Table 2 (Reddit Post, two completions), $U=5$ performs better than $U=3$. This is likely because the small preference data size on the server makes it easier for the LLM to "cheat" all three selectors by producing superficially favored responses without genuine improvement. This is an issue we refer to as reward hacking. By increasing $U$, the system becomes more robust to such gaming behavior but at the cost of higher complexity.
>
> In summary, $U$ must balance heterogeneity handling, computational feasibility, and robustness to reward hacking. Future work could further explore adaptive strategies to dynamically adjust $U$ based on task-specific needs and data distribution.

---

> > ### Author Response · Authors · 2024-11-20
> > **Official Comment by Authors (2/2)**
> >
> > > **[W3]** The additional complexity introduced by clustering and the use of multiple selectors in FedBiscuit may still pose scalability challenges, particularly in real-world applications with thousands of clients. It would be helpful to provide more analysis on the scalability of the approach, especially under varying numbers of clients and communication costs. This would shed light on the method’s feasibility for large-scale deployment.
> >
> > Thanks for your insightful comments. We appreciate your interest in understanding the scalability of FedBiscuit for large-scale deployments. In this work, we freeze the pretrained LLM, which is distributed to all clients at the beginning of each training round. Regarding training, the total communication cost is $O(AR \cdot C)$ in FedBis and $O((AR+MU\lfloor R/\tau \rfloor) \cdot C)$ in FedBiscuit, where $A$ is the number of sampled clients at each round, $M$ is the number of total clients, $C$ is communication cost of a binary selector, $R$ is the total number of rounds, and $\tau$ is the frequency of full-client participation.
> >
> > While periodic full-client participation may introduce scalability challenges, it is a practical approach widely used in federated learning literature [1, 2]. Such methods have been shown to improve convergence efficiency both theoretically and empirically. Importantly, FedBiscuit incorporates partial client participation, significantly enhancing its scalability and making it suitable for real-world applications involving thousands of clients. Unlike most existing works [3, 4, 5, 6, 7], which assume full-client participation, FedBiscuit explicitly supports partial client participation, making it one of the few approaches designed for large-scale federated learning deployment.
> >
> >
> > **References:**
> > [1] FedPAGE: A Fast Local Stochastic Gradient Method for Communication-Efficient Federated Learning
> > [2] MARINA: Faster Non-Convex Distributed Learning with Compression
> > [3] FedBiOT: LLM Local Fine-tuning in Federated Learning without Full Model
> > [4] FederatedScopeLLM: A Comprehensive Package for Fine-tuning Large Language Models in Federated Learning
> > [5] Efficient federated prompt tuning for black-box large pre-trained models
> > [6] FedBPT: Efficient Federated Black-box Prompt Tuning for Large Language Models
> > [7] Towards Building the Federated GPT: Federated Instruction Tuning

---

> ### Comment · Reviewer_EwYt · 2024-11-22
> **Thanks**
>
> Thanks for the explanations provided. It has clarified my questions and I will keep the score.

---

> > ### Author Response · Authors · 2024-11-22
> >
> > Dear Reviewer EwYt,
> >
> > Thank you for your kind acknowledgment. We are happy to hear that your questions have been addressed, and we truly appreciate your positive feedback and rating.
> >
> > Best,
> >
> > Authors

---

> > ### Author Response · Authors · 2024-12-02
> >
> > Dear Reviewer EwYt,
> >
> > Thank you for your prompt reply and for finding our responses to your concerns satisfactory.
> >
> > We sincerely appreciate your efforts in providing feedback on our work, which has significantly improved the quality of our work. Given that our responses have addressed your questions and weaknesses, we hope you will consider adjusting the rating to reflect this resolution if you find it appropriate.
> >
> > Thank you once again for your time and effort in reviewing our submission.
> >
> > Best,
> >
> > Authors

---

### Official Review · Reviewer_9Rrf · 2024-11-05

**Soundness:** 3
**Presentation:** 4
**Contribution:** 3
**Rating:** 5
**Confidence:** 3

**Summary:**

The paper deals with privacy preservation through applying federated learning to employing user preferneces to fine-tune a common LLM. The synergy of reinforcement learning, large language model, and federated learning appears apealing to me. The paper presents nicely with an intuitive comparison between standard and fedrated RLHF, making the problem easy to follow.

**Strengths:**

- The paper deals with an important research problem.
- The paper is generally well orgnaised and presente, and easy to follow.
- The authors proposed a sound solution to the research problem, showing advantages over baselines.

**Weaknesses:**

- The challenged identifiied on Page 2 seems univeral and apply to many federated learning scenarios, not specified to this paper's research context. Therefore, although the overall problem setting seems to be new, the key research problem to tackle remains conventional.
- To deal with excessive comptuation overhad and preference heterogeneity seem to be a common issue that has been addressed by various previous efforts. The clustering approach to addressing the key issue does not appear novel to me.
- The cons of appplying the federated learning approach might be discussed for readers to better understand the trade-off.

**Questions:**

What are the unique challenges posed by employing federated learning to RLHF?
How RLHF impact the challenges and solution design presented in the paper?
What is the motivation behind the binary selector? what are the alternatives and why binary selectior, in particular, is chosen to be part of the solution?

---

> ### Author Response · Authors · 2024-11-20
> **Official Comment by Authors (1/2)**
>
> > **[W1]** The challenges identified on Page 2 seems universal and apply to many federated learning scenarios, not specified to this paper's research context. Therefore, although the overall problem setting seems to be new, the key research problem to tackle remains conventional.
>
> Thank you for your insightful comments. While we agree that some challenges in federated learning (FL) are universal, our work introduces intrinsic differences specific to federated RLHF, distinguishing it from conventional FL:
>
> - **Computation Overhead:** Traditional FL typically optimizes a model using standard one-to-one mappings $(x, y)$, where $x$ is the input and $y$ is the groundtruth. In contrast, FedRLHF optimizes a reward model using pairwise preference data $(x, y_w, y_l)$, ensuring that the preferred completion $(x, y_w)$ scores higher than the dispreferred one $(x, y_l)$. This results in maintaining two computation graphs -- one for each preference -- during optimization, whereas traditional FL only requires a single graph. This distinction introduces a unique computational challenge.
> - **Data Heterogeneity:** While both traditional FL and our work face data heterogeneity due to clients’ diverse data distributions, our work uniquely considers *preference heterogeneity*. Given two completions $y_0$ and $y_1$ for an input $x$, client preferences can differ -- some may prefer $y_0$, while others prefer $y_1$ [1, 2, 3]. This leads to a fundamentally different objective: unlike traditional FL, which trains models for tasks with clear, objective answers (e.g., classification), our approach aligns a large language model (LLM) with the majority’s subjective preferences.
>
> These distinctions are pivotal to our research and highlight why the challenges and solutions proposed are tailored specifically to federated RLHF.
>
> **References:**
> [1] Personalized Soups: Personalized Large Language Model Alignment via Post-hoc Parameter Merging
> [2] MaxMin-RLHF: Towards Equitable Alignment of Large Language Models with Diverse Human Preferences
> [3] AlpacaFarm: A Simulation Framework for Methods that Learn from Human Feedback
>
> > **[W2]** To deal with excessive comptuation overhead and preference heterogeneity seem to be a common issue that has been addressed by various previous efforts. The clustering approach to addressing the key issue does not appear novel to me.
>
> Thanks for your comments. As outlined in the response to **[W1]**, the causes of excessive computation overhead and preference heterogeneity in our work differ fundamentally from those in traditional FL. To this end, we apply two techniques to address these challenges:
>
> - **Binary Selector:** This technique removes the extra computation graph inherent in federated RLHF, effectively reducing computational costs while maintaining performance. Unlike conventional FL models that rely on a single graph, federated RLHF requires managing two computation graphs, a problem that our binary selector resolves effectively.
> - **Clustering Approach:** While clustering has been explored in existing FL literature [1, 2], our integration is far from straightforward. Traditional algorithms predetermine the number of models and partition clients into groups, with each group training its own model. This one-to-one mapping assumes a fixed number of groups, but in practice, predefining the number of groups is challenging. Untrained selectors emerge when no clients are assigned to certain groups, introducing significant risks: Excluding untrained selectors simplifies reward hacking, as the LLM may learn to "cheat" the system by producing favored responses without genuine improvement. Conversely, including untrained selectors risks misalignment, leading to incorrect preference alignments.
>
> To address these challenges, we propose a sustainable algorithm ensuring all selectors contribute meaningfully, mitigating reward hacking while maintaining the accurate preference reflection of most clients.
>
> **References:**
> [1] Clustered federated learning: Model-agnostic distributed multitask optimization under privacy constraints
> [2] Structured federated learning through clustered additive modeling

---

> > ### Author Response · Authors · 2024-11-20
> > **Official Comment by Authors (2/2)**
> >
> > > **[W3]** The cons of applying the federated learning approach might be discussed for readers to better understand the trade-off.
> >
> > Thanks for your question. If we understand correctly, you are referring to the trade-off between performance and privacy when applying FL. Compared to centralized methods, FL may result in a modest performance drop due to limited access to global data and the inherent challenges of model aggregation across heterogeneous client data. However, this trade-off is necessary to **safeguard user privacy**, which is a critical consideration in real-world applications. In other words, FL is indispensable for scenarios where user **data confidentiality** is paramount and centralized learning is impractical or non-compliant with privacy regulations. Our work aims to maximize the benefits of FL while mitigating its performance drop through a clustering approach, making it viable for practical deployment.
> >
> > > **[Q]** What are the unique challenges posed by employing federated learning to RLHF?
> >
> > Thanks for your question. Applying FL to RLHF introduces several unique challenges: Unlike traditional RLHF, which assumes centralized, consensus-based preference labels, FL involves decentralized clients with potentially inconsistent preferences, complicating the training of a unified reward model. Additionally, while traditional RLHF leverages centralized entities with abundant computational resources, FL operates under the constraint of limited local computation on client devices, seeking resource-efficient methods for training. For further details, please refer to our response to **[W1]**.
> >
> > > **[Q]** How RLHF impact the challenges and solution design presented in the paper?
> >
> > Thanks for your question. RLHF introduces specific challenges and directly impacts our solution design in the following ways:
> >
> > - **Preference Data and Model Design:** RLHF relies on preference data, which includes an input paired with a preferred and a dispreferred completion. To handle this, we designed a model, a binary selector, capable of determining which completion is better within a given pair. This ensures that the selector is effectively trained on preference-based feedback.
> > - **Reward Hacking Mitigation:** A common challenge in RLHF is *reward hacking*, where the LLM optimizes for misleading feedback rather than genuine improvement. To address this, we employ multiple binary selectors. This ensemble approach mitigates reward hacking by diversifying the evaluation criteria, making it harder for the LLM to exploit any single selector’s bias.
> >
> > > **[Q]** What is the motivation behind the binary selector? What are the alternatives and why binary selectior, in particular, is chosen to be part of the solution?
> >
> > Thanks for your questions. The motivation behind the binary selector is attributed to the need to address excessive computational overhead during local training. As discussed in **[W1]**, it involves maintaining two computation graphs when applying traditional RLHF approaches for reward model training. An intuitive way to reduce computational complexity is to reformulate the task as a **classification problem**, where the model selects the better completion from a given pair. This leverages recent advancements in LLMs, which have demonstrated superior performance in multi-choice tasks. By fine-tuning a pretrained LLM, we enable it to make pairwise comparisons and choose the preferred completion efficiently.
> >
> > In addition to improving computation efficiency, the binary selector allows the LLM to make a holistic comparison between the two completions, leveraging its contextual understanding to provide a more accurate and reliable preference. In contrast, the traditional RLHF approach may fail to capture nuanced preferences, leading to suboptimal comparisons, because it separately grades each completion with a reward model and compares their scores.

---

> > > ### Author Response · Authors · 2024-11-24
> > > **We look forward to your acknowledgement**
> > >
> > > Dear Reviewer 9Rrf,
> > >
> > > Thank you for your time and insightful comments on our work. As we approach the end of the author-reviewer discussion period, we greatly value the opportunity to address your concerns.
> > >
> > > Could you kindly review our responses to see if they address your comments? We would highly appreciate it if you find our responses satisfactory and consider updating your rating. Feel free to reach out if you have any other questions or need further clarification.
> > >
> > > Best,
> > >
> > > Authors

---

> > > ### Author Response · Authors · 2024-11-27
> > > **We look forward to your follow-up comments**
> > >
> > > Dear Reviewer 9Rrf,
> > >
> > > Thanks a lot for your commitment to reviewing our paper and providing insightful feedback. We hope this message finds you well.
> > >
> > > We would like to kindly remind you that we have less than 24 hours remaining to finalize our manuscript. Based on your valuable feedback, we have thoroughly revised our paper, and the updated content has been highlighted in red. Moreover, we have provided detailed responses to your comments. We are eager to hear your feedback to see if our revisions and responses have fully addressed your concerns, and we hope you might consider increasing your rating if the changes meet your expectations.
> > >
> > > If you have any further suggestions or additional concerns, please do not hesitate to let us know. We deeply value your comments and will try to address any remaining points.
> > >
> > > Thank you once again for your time and efforts.
> > >
> > > Best,
> > >
> > > Authors

---

> ### Author Response · Authors · 2024-12-03
> **Gentle Reminder for Discussion**
>
> Dear Reviewer 9Rrf,
>
> We sincerely appreciate your time and efforts in reviewing our paper and providing valuable comments and suggestions. Based on your insightful feedback, we have revised our paper and addressed your concerns in our rebuttal. A summary of our responses is as follows:
>
> - **Differences between federated RLHF and traditional federated learning (FL):** Federated RLHF introduces unique challenges that do not present in traditional FL: (1) Computation overhead due to the pairwise nature of preference data, requiring two computation graphs for optimization; and (2) Preference heterogeneity, where clients may have conflicting preferences for the same pair of model completions. These differences motivate a specialized solution to federated RLHF.
> - **Motivations of binary selector:** The binary selector addresses excessive computation overhead by reformulating the task as a binary classification problem, enabling efficient pairwise comparisons. This approach leverages recent advances in LLMs for multi-choice tasks, allowing the model to compare pairs of completions holistically.
> - **Motivations of the proposed clustering approach in FedBiscuit:** This clustering approach effectively groups clients with similar preferences, addressing preference heterogeneity and reflecting preferences across diverse clients. Moreover, it returns multiple well-trained binary selectors, mitigating reward hacking because the LLM should generate the completions satisfied by most selectors.
> - **Trade-off between performance and privacy:** While FL may incur a modest performance drop compared to centralized methods, it is indispensable for scenarios requiring stringent data confidentiality. Our clustering and binary selector approaches mitigate the trade-off, making federated RLHF viable for real-world applications.
>
> For more details, please feel free to take a look at the above **Official Comment by Authors**.
>
> As the discussion will end in **8 hours**, we would like to kindly remind you to review our revisions and rebuttal and provide feedback on whether our revisions and rebuttal address your concerns. If you have any additional questions/concerns/suggestions, please do not hesitate to let us know. We will do our best to respond promptly and address them.
>
> Best,
>
> Authors of **Paper 13225 (Towards Federated RLHF with Aggregated Client Preference for LLMs)**

---

### Official Review · Reviewer_t8Tn · 2024-11-11

**Soundness:** 2
**Presentation:** 2
**Contribution:** 3
**Rating:** 6
**Confidence:** 3

**Summary:**

Reinforcement learning with human feedback (RLHF) is a good way to fine-tune a pretrained large language model (LLM) to generate content aligned with human preferences. This paper proposes to use federated learning technique to aggregate different model information from users to enhance RLHF ability of LLM. Concretely, the proposed federated RLHF methods encode each client’s preferences into binary selectors and aggregate them to capture common preferences. This method overcomes key challenges like preference heterogeneity and reward hacking to enhance LLM output quality. Experiments on two LLM benchmarks demonstrate the effectiveness of the proposed federated RLHF algorithm.

**Strengths:**

(1)	This paper provides a novel federated learning-based reinforcement learning method to enhance the output ability of LLM. To the best of my knowledge, this is the first time to employ federated learning technique to enable diverse user collection for RLHF.

(2)	The organization of this paper is clear and easy to follow. In particular, Figure 2 clearly depicts the outline of the proposed FedBis model.

(3)	The algorithm design in Section 5.1 is practical and new to me. Besides, the key component of the algorithm (i.e. training a binary selector and aggregating different selectors to the LLM server) effectively incorporates federated learning idea into the design of RLHF model.

(4)	The experimental results on summarization and question-answering dataset validate the advantages of the proposed algorithm over existing methods.

**Weaknesses:**

(1)	The experimental results in Section 7 are not extensive enough. This paper conducts experiments on two NLP tasks, summarization and question-answering, and more experiments on other tasks should be complemented, like few-shot learning, synthetic, code completion, multi-needle retrieval.

(2)	I admire the proposed FL-based reinforcement learning method, but it will be more readable and more concise to summarize the contents in Section 5.1 Algorithm Design into a pseudocode (i.e. shown in an Algorithm).

(3)	Lines 70-77 in Introduction is about the limitation of preference heterogeneity and reward hacking, and such limitation appears (in detail) once again in lines 224-237 of methodology. It suffices to mention such limitation once in one paper, or at least, not to emphasize it twice (e.g. not use bold to highlight it).

(4)	Section 6 is about the details of the experimental datasets, and will be better to defer it to Section 7 Experiments.

(5)	It will be better to use \begin{small} or other writing skill, to make Eq.(2) and Eq.(4) shown in one line.

(6)	Lines 237,294, 471, isolated word

(7)	Line 268, fix “. $\phi _{u}$”

**Questions:**

It may not be easy to add more experimental results in the rebuttal period. But can the proposed method also enhance RLHF ability of LLM in other NLP tasks like few-shot learning, synthetic, code completion, multi-needle retrieval?

---

> ### Author Response · Authors · 2024-11-20
>
> > **[W1]** The experimental results in Section 7 are not extensive enough. This paper conducts experiments on two NLP tasks, summarization and question-answering, and more experiments on other tasks should be complemented, like few-shot learning, synthetic, code completion, multi-needle retrieval.
> > **[Q1]** It may not be easy to add more experimental results in the rebuttal period. But can the proposed method also enhance RLHF ability of LLM in other NLP tasks like few-shot learning, synthetic, code completion, multi-needle retrieval?
>
> Thanks for your suggestion. The datasets of federated RLHF should satisfy two conditions: (i) the data can be non-i.i.d. partitioned, and (ii) the data should be in the form of an input along with preferred and dispreferred completions. To the best of our knowledge, there are two public datasets that satisfy both requirements, i.e., Reddit TLDR and SHP datasets. Reddit TLDR is a *summarization* dataset that can be partitioned according to the worker ID, while SHP is a *question-answering* dataset that can be partitioned by the question domains.
>
> Our work aims to solve general NLP tasks in open-ended text generation, and the experimental results demonstrate the superiority of FedBis and FedBiscuit over the existing baselines, thereby validating the effectiveness of the proposed methods. The mentioned tasks (e.g., few-shot learning, synthetic, code completion, multi-needle retrieval) could act as extra constraints on top of the current federated RLHF scenario and introduce more challenging, meaningful research topics. These topics will be explored in future studies. As the proposed FedBis and FedBiscuit are the general solutions to federated RLHF, they can be extended to the mentioned topics as baselines.
>
> > **[W2]** I admire the proposed FL-based reinforcement learning method, but it will be more readable and more concise to summarize the contents in Section 5.1 Algorithm Design into a pseudocode (i.e. shown in an Algorithm).
> > **[W3]** Lines 70-77 in Introduction is about the limitation of preference heterogeneity and reward hacking, and such limitation appears (in detail) once again in lines 224-237 of methodology. It suffices to mention such limitation once in one paper, or at least, not to emphasize it twice (e.g. not use bold to highlight it).
> > **[W5]** It will be better to use \begin{small} or other writing skill, to make Eq.(2) and Eq.(4) shown in one line.
> > **[W6]** Lines 237,294, 471, isolated word
> > **[W7]** Line 268, fix "$\phi_u$."
>
> Thanks for your suggestion. In our revised manuscript, we fix the issues that you mentioned, and we highlight them in red.
>
> > **[W4]** Section 6 is about the details of the experimental datasets, and will be better to defer it to Section 7 Experiments.
>
> Thank you for your suggestion. Section 6 introduces two federated human preference datasets, and since this is the first work to propose a preference dataset partition in federated learning, we believe it is important to highlight them in a separate section. This allows us to emphasize the novelty and significance of our contribution before presenting the experimental results in Section 7.

---

> > ### Author Response · Authors · 2024-11-24
> > **We look forward to your acknowledgement**
> >
> > Dear Reviewer t8Tn,
> >
> > Thank you for your time and insightful comments on our work. As we approach the end of the author-reviewer discussion period, we greatly value the opportunity to address your concerns.
> >
> > Could you kindly review our responses to see if they address your comments? We would highly appreciate it if you find our responses satisfactory and consider updating your rating. Feel free to reach out if you have any other questions or need further clarification.
> >
> > Best,
> >
> > Authors

---

> > ### Author Response · Authors · 2024-11-27
> > **We look forward to your follow-up comments**
> >
> > Dear Reviewer t8Tn,
> >
> > Thanks a lot for your commitment to reviewing our paper and providing insightful feedback. We hope this message finds you well.
> >
> > We would like to kindly remind you that we have less than 24 hours remaining to finalize our manuscript. Based on your valuable feedback, we have thoroughly revised our paper, and the updated content has been highlighted in red. Moreover, we have provided detailed responses to your comments. We are eager to hear your feedback to see if our revisions and responses have fully addressed your concerns, and we hope you might consider increasing your rating if the changes meet your expectations.
> >
> > If you have any further suggestions or additional concerns, please do not hesitate to let us know. We deeply value your comments and will try to address any remaining points.
> >
> > Thank you once again for your time and efforts.
> >
> > Best,
> >
> > Authors

---

> ### Author Response · Authors · 2024-12-03
> **Gentle Reminder for Discussion**
>
> Dear Reviewer t8Tn,
>
> We sincerely appreciate your time and efforts in reviewing our paper and providing valuable comments and suggestions. Based on your insightful feedback, we have revised our paper and addressed your concerns in our rebuttal. A summary of our rebuttal is as follows:
>
> - **NLP tasks in experiments:** In the context of public RLHF datasets, only Reddit TLDR and SHP datasets support non-i.i.d. data partitioning. Our experiments utilize these two datasets to demonstrate the performance and superiority of our proposed methods (FedBis and FedBiscuit) across two general NLP tasks: summarization and question-answering. While the mentioned tasks (e.g., few-shot learning, synthetic data, code completion) introduce more challenging constraints, they fall within the broader scope of general NLP tasks and present meaningful directions for future research.
> - **Revisions based on your feedback:** We have thoroughly addressed your suggestions, including improving the readability of algorithms, resolving isolated word and equation formatting issues, and avoiding redundant content in the manuscript. These changes have been highlighted in red in the revised version for easy reference.
> - **Contribution regarding non-i.i.d. partition on a public RLHF dataset:** Since this is the first work that introduces how to partition a public RLHF dataset in a non-i.i.d. manner, we made the relevant content a standalone section to better emphasize this contribution.
>
> For more details, please feel free to take a look at the above **Official Comment by Authors**.
>
> As the discussion will end in **8 hours**, we would like to kindly remind you to review our revisions and rebuttal and provide feedback on whether our revisions and rebuttal address your concerns. If you have any additional questions/concerns/suggestions, please do not hesitate to let us know. We will do our best to respond promptly and address them.
>
> Best,
>
> Authors of **Paper 13225 (Towards Federated RLHF with Aggregated Client Preference for LLMs)**

---

### Meta-Review · Area_Chair_y2Nt · 2024-12-21

**Metareview:**

Reinforcement learning with human feedback (RLHF) is an effective method for fine-tuning pretrained large language models (LLMs) to align their outputs with human preferences. This paper introduces a novel approach that leverages federated learning to aggregate diverse model information from users, enhancing the RLHF capabilities of LLMs. Specifically, the proposed federated RLHF framework encodes each client’s preferences into binary selectors, which are aggregated to identify and capture shared preferences across users. This approach addresses critical challenges such as preference heterogeneity and reward hacking, significantly improving the quality of LLM outputs. Experimental results on two LLM benchmarks validate the effectiveness of the proposed federated RLHF algorithm.

+ The paper introduced an interesting model for federated learning
+ The paper is well written
- The computational cost of the model can be further analyzed
- The experiments can be further improved by introducing more baselines

**Additional Comments On Reviewer Discussion:**

In the rebuttal period, the authors have well addressed the reviewers' concerns. I believe the comments of the reviewer who gives  5 can be easily revised in the final version. I think the paper can be accepted.

---

### Decision · Program_Chairs · 2025-01-22

Accept (Poster)